# Complex networks applied to political analysis: Group voting behavior in the Brazilian congress

**Tiago José de Oliveira Toledo Junior**[ID]*,
**Diego Raphael Amancio, Roseli Aparecida Francelin Romero**

Institute of Mathematics and Computer Science, University of São Paulo, São Carlos, Brazil

* tiago.toledo@usp.br (TTJ)

**Data availability statement:** All files are available in a GitHub repository at https://github.com/TNanukem/complex_networks_political_analysis.

## Abstract

The Senate and the Chamber of Deputies constitute the Brazilian Congress and are responsible for the Brazilian legislative management. Complex networks were shown to be a suitable tool to analyze this type of system. Several researches explored party dynamics in the Chamber of Deputies, however, no attention has been given to the Senate. Previous works that have stated the necessity of a backbone extraction methodology to be used in these types of networks also failed to define an automatic backbone extraction methodology to uncover group structure in legislative networks, reverting to heuristics or subjective approaches. In this work, we explore both legislative houses and compare them to see their differences and similarities. We also systematize an automatic backbone extraction methodology. Further, we expand on previous analyses by bringing spectrum and government x opposition analysis based on voting data. Our results show that the Senate and the Chamber of Deputies have behaved differently during major events in Brazil over the second decade of the century. From the obtained results it is fair to say that the dynamics for both houses are different and that the best backbone extraction algorithm varies over time and is different for each house.

## Introduction

The growth of data availability and data science tools to analyze large amounts of data has generated advances in several areas of knowledge. Some of them are considered complex systems, in which traditional tools are not the most viable option to perform data analysis due to the higher complexity of the system [1]. In this context, an important framework that has gained considerable attention is the representation of complex networks. Its utility has been extensively demonstrated in various applications in various domains such as Social Networks [2], Economy [3], Computer Science [4], Medicine [5] and several others [6–8]. For an extensive overview of Complex Networks applications in different areas, we refer to [9].

The political dynamics of a country can be considered a complex system. This type of system comprises economic results, law dynamics, justice decisions, and executive decisions by the elected president and governors. Then we can say that a country comprises several smaller complex systems. The present work focuses on the federal legislative system in Brazil.

**Funding:** Funding provided by São Paulo Research Foundation (FAPESP) for the publication of the research.

**Competing interests:** The authors have declared that no competing interests exist.

Several works tried to use complex networks and social network analysis to study the social aspects of political affairs [1,10–16] since politics in democratic countries is necessarily a social endeavor [12].

Two houses constitute the Brazilian legislative system, the lower house is called the Chamber of Deputies, and the higher house is called the Senate. The work proposed by [1,11,16,17] used complex networks and social networks to analyze only the Chamber of Deputies, the largest Brazilian house.

It does not matter in which house law is proposed first, before it goes for presidential sanction, relying on the Chamber of Deputies only exploring half of the legislative picture.

Some works have stated the necessity of applying backbone extraction methodologies to improve the performance of analysis of political networks. However, only the work defined by [15] defined a model selection methodology and no work explored the possibility of having strategies for a network at several points in time.

Therefore, the main research question is to understand if the party behavior is the same for the Senate and the Chamber of Deputies and if they can be analyzed using the same set of tools. The specific objectives aiming to answer the main research question are:

- Verify if the best backbone extraction algorithm is the same for both houses and if it is the same at every period.
- Verify if the party structure maximizes the modularity or the surprise of the network in the Senate, or if the community detection methodology already shown to maximize modularity in the Chamber of Deputies is also applicable here. Furthermore, we want to verify if there are similarities between the communities in both houses.
- Verify if the parties' behavior, in terms of fragmentation and isolation, is the same in both houses.

The main contributions of this work are:

- We expand previous works to analyze the Senate and compare its characteristics to the Chamber of Deputies, covering the entire federal legislative system for the first time.
- We show that the methodology most adequate for the Chamber of Deputies is able to uncover hidden group structures for the Senate.
- We prove that political parties are not the group definition that maximizes the modularity and the Surprise of the Senate networks and that the number of effective groups is smaller than the number of parties.
- We systematize an automatic backbone extraction methodology to select the most adequate methodology for each network in our study, focusing on information retrieval of the selected methods improving over previous research that had non-data-driven choices of backbone extraction algorithms [1,11].
- We explore the political spectrum of the parties, showing how parties from the same political spectrum behave and show that the behavior of the spectrum differs according to the house.
- We define a data-driven government and opposition definition, based on voting data and not on announcements from the parties, and analyze the houses based on this classification.

The data and code for this study are available at https://github.com/TNanukem/complex_networks_political_analysis.

## Related works

### Complex networks applied to political analysis

Political parties provide smaller modularity as compared to communities generated with community detection algorithms. The government coalition is more cohesive than the opposition coalition. These results were generated for the Italian Parliament by [13]. In this analysis, a voting network, for the period between April and December of 2013, was constructed in which the political party cohesion was measured by verifying the intra and inter-cluster densities. Therefore, there is a number of deputies who are not loyal to their parties in terms of voting behavior. Further, a two-community structure was found, representing roughly the government and opposition.

In [1], it was also shown that a detected community structure has higher modularity than the political parties, in this case for the Brazilian Chamber of Deputies. Their research explored the characterization of Brazilian political parties in terms of isolation, fragmentation, and coalition with other parties based on how similar the voting patterns of the deputies of these parties are. Yearly networks were built using the number of agreed deputies' votes normalized by the number of votes in that year. A backbone extraction mechanism named Disparity Filter [18] was used to extract the network backbones before applying the Leiden [19] algorithm to uncover the communities. They also showed a strong isolation of the governing party in the moments preceding the Impeachment process held in 2015 and 2016 in Brazil.

Party cohesion is variable per party and depends on whether the party is in the majority or minority and whether it makes part of the government or the opposition. This was found for the Canadian House of Commons by [14], where the voting network was studied in terms of cohesion and discipline. The alignments of the partisans were measured by visual inspection of the network plot, generated using the ForceAtlas2 algorithm [20].

The structural balance of the Brazilian Deputies Chamber was analyzed in [17]. A signed network was generated using the voting agreement deputies. Then a clustering algorithm called ILS-CC was used to discover groups among deputies. The resulting clusters showed the breakdown of the political coalition that led to President Dilma Rousseff reelection in 2014 right before her impeachment process was concluded. It was also shown that deputies actuated as mediators who worked to align the government and the opposition.

The Brazilian Deputies Chamber compared to the US Congress presents a higher variability in ideological behavior. On the other hand, it has a smaller set of ideological communities as compared to the actual number of political parties. These were the findings found in [11], which generated a network from the deputies voting data, and generated communities using the Louvain [21] algorithm. The networks are generated using the ratio of equal votes between two deputies concerning the total number of votes to define edge weights. These edge weights were then pruned from the resulting network to remove every edge with a weight below the 90th percentile.

The work presented in [22] analyzes the stability of the Brazilian Chamber of Deputies by creating a network based on the correlation between votes. Then, they apply a Minimum Spanning Tree (MST) and show that political terms usually start with strong opposition and converge for a consensus by the end of the term. They also verified a breakdown of the coalition government and isolation of the government party preceding the impeachment process.

The political instability of Dilma Rousseff's government started as soon as her first mandate [23]. This study also states that the Brazilian Deputies Chamber can be represented by 2 groups during political stability, with periods of instability presenting more effective groups.

## Backbone extraction methodologies

A framework for co-interaction analysis systems between several agents, such as voting networks for political systems that were proposed in [15]. The same framework was implemented in [1,11]. He also proposes a framework to select the best backbone extraction methodology for the problem at hand using two types of analysis: a topological consisting of network metrics, and a contextual analysis, consisting of a regression model to recover information from the network after the backbone process.

To overcome the limitations of filtering strategies for network edge reduction, [18] proposed a backbone extraction methodology named by disparity filter. The idea is to consider local fluctuations of the edge weights of a node to estimate which ones are relevant. Given the normalized weights for each edge on each node, a null hypothesis is constructed stating that the edges for a node follow a uniform distribution of weights. Then, a p-value is calculated for each edge to verify the alternative hypothesis that the edge does not follow the null model. When the null hypothesis is rejected, the edge is considered important. Therefore, it is kept in the network. They have shown that this method is better at keeping the network structure, especially for highly skewed networks compared to the threshold filtering method because it keeps more of the total weight of the network.

The backbone extraction methodology proposed by [24], called Locally Adaptive Network Sparsification, aims to be a non-parametric model, making no assumption about the underlying edge weight distribution and instead using the empirical distribution found in the network. To do so, the normalized edge weight is calculated. Then, the cumulative distribution function for the normalized weight is calculated for each node. A significance level is defined, and all edges having a probability smaller than it are kept. The authors state that this method is better at preserving the structure of networks where the underlying weight distribution has high heterogeneity.

The concept of salience can also be used to extract the backbone of a network. Salience is defined as the fraction of the tree's shortest-path for all nodes in the network containing a given edge. In [25], it was shown that this metric has a bimodal distribution around the values of 0 and 1. Therefore, it can be used as a binary classifier of important edges.

To improve the performance of the Disparity Filter algorithm, because it takes into account only one of the edge nodes, the Noise Corrected algorithm was proposed in [26]. The method assumes that the edge weights are drawn from a binomial distribution. Then, a Bayesian method is used to estimate the probability of finding an edge of that given weight between those two nodes. The edge is dropped if its weight is less than $\delta$ standard deviations from the expectation.

A two-step algorithm can also extract the backbone from a network. This is the main idea behind the Doubly Stochastic Filter proposed by [27]. The first step transforms the network adjacency matrix into a doubly stochastic matrix. Then, edges are added sequentially, the greater ones first, until all nodes are included in a single connected component.

The Netbone library [28] provides the implementation of a suite of backbone extraction methodologies, some of which were used in the present work and described in the Methodology section. All of the implementations used in this work came from this library. It also provides filtering methods that automatically select the cutoff points for its implemented methods based on some criteria. Some of the filters considered are the threshold filter, which extracts the network based on a given p-value or score threshold, and the rate of a given percentage of the original edges in the extracted network.

## Methodology

The methodology consists of four steps illustrated in Fig 1.

The first step consists of creating a network representation from raw voting data. This process usually generates overly dense networks filled with noisy edges [1], since not all Congress sessions are discriminative of opinion or ideology [11]. A backbone extraction methodology is applied to deal with noisy edges aiming to uncover the backbone structure that represents the phenomena we are interested in analyzing.

To improve over previous research [1,11,13], which have not use backbone extraction algorithm and nor have chosen the algorithms arbitrarily, we use a pipeline for the selection of the algorithm based on [15]. The idea is to compare methodologies based on performance metrics to verify whether the resulting network is still able to identify the phenomenon of our interest, which consists of a contextual analysis.

Once the backbone extraction algorithm is selected, a community detection algorithm is applied to identify the ideological communities of the network. Finally, these communities and the overall networks are characterized and analyzed to uncover relationships and patterns in the data.

The selected period for the analysis was from 2010 (inclusive) to 2021 (inclusive).

### Network construction

The network is built using a split of one year. For this, a network per year will be created for each house, totaling 24 networks. For each network, the nodes are members of a determined house.

We adopt the following process to define the weight of each edge. The similarity between the nodes is based on counting the number of agreed and disagreed votes between the

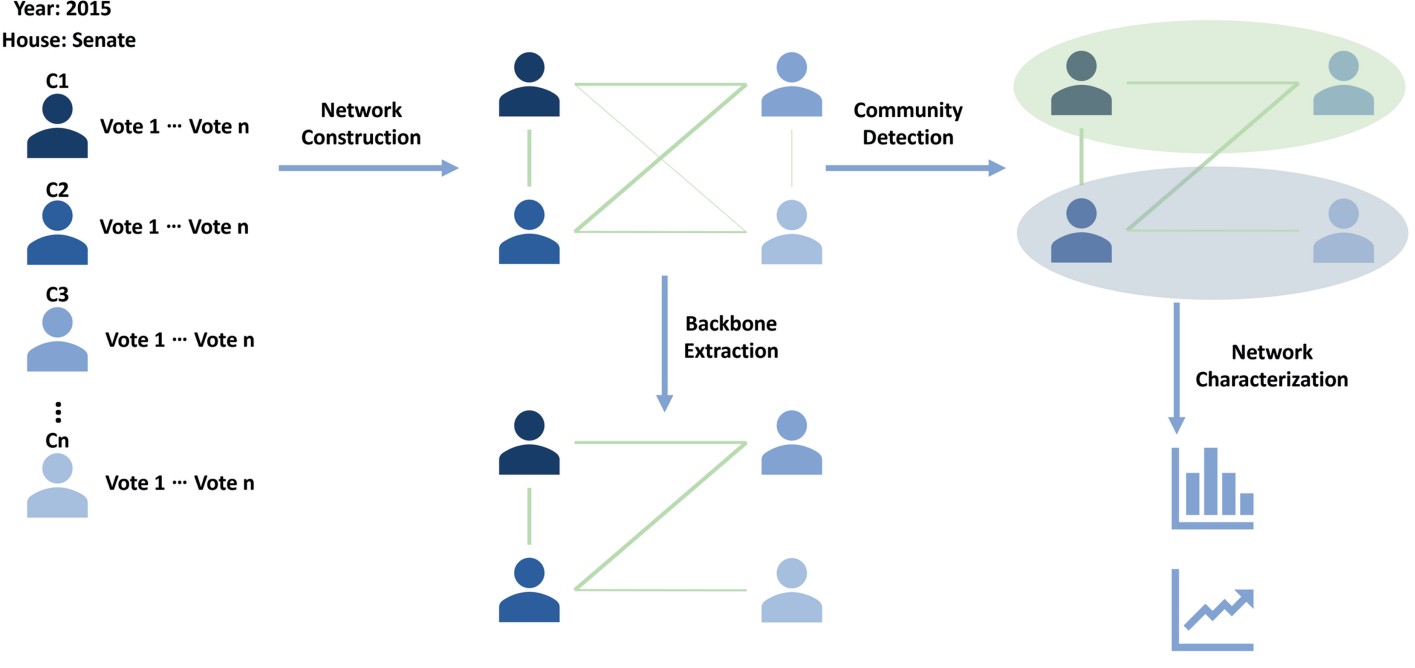

**Fig 1. Methodology diagram.**

deputies. Specifically, if two deputies voted the same way in a voting session, their edge weight was increased by 1. Conversely, if they voted differently, the edge weight value was decreased by 1. If one of the deputies was absent during the voting session, there is no change in edge weight. Finally, the edge weight values are normalized by the total number of voting sessions in the respective year. Only edges with a weight value greater than zero are kept.

To keep the consistency between the two houses of the Brazilian Congress, which can have varying types of voting mechanisms (i.e. the Senate sometimes employs anonymous voting), our study limited the scope of analysis to non-anonymous sessions. Specifically, we only considered votes categorized as "yes", "no", "abstention" and "obstruction".

**Network construction example.** Consider four deputies who have voted in 4 sessions with the following votes:

- **Deputy 1** - ['Yes', 'Yes', 'No', 'NaN']
- **Deputy 2**- ['No', 'No', 'Yes', 'Yes']
- **Deputy 3** - ['Yes', 'No', 'No', 'Abstention']
- **Deputy 4** - ['Yes', 'NaN', 'Yes', 'Abstention']

in which 'NaN' stands for a deputy not voting in the session (which is different from abstaining). In this case, the agreement vector between deputies is:

- **Agreement 1-2** = [-1, -1, -1, 0]
- **Agreement 1-3** = [1, -1, 1, 0]
- **Agreement 1-4** = [0, 0, -1, 0]
- **Agreement 2-3** = [-1, 1, -1, -1]
- **Agreement 2-4** = [0, 0, 1, -1]
- **Agreement 3-4** = [1, 0, -1, 1]

Summing those values and normalizing by the number of sections would generate the network presented in Fig 2.

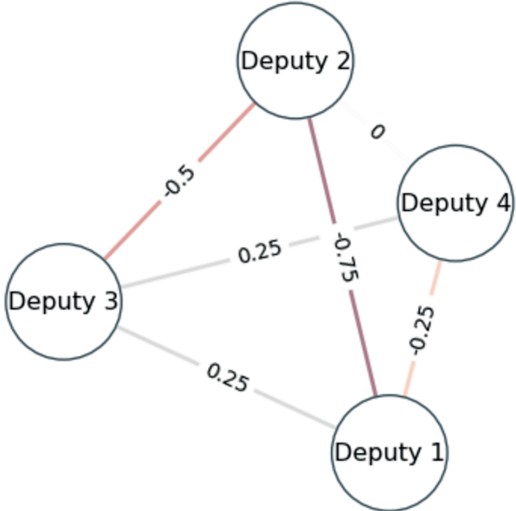

**Fig 2. Example voting network.**

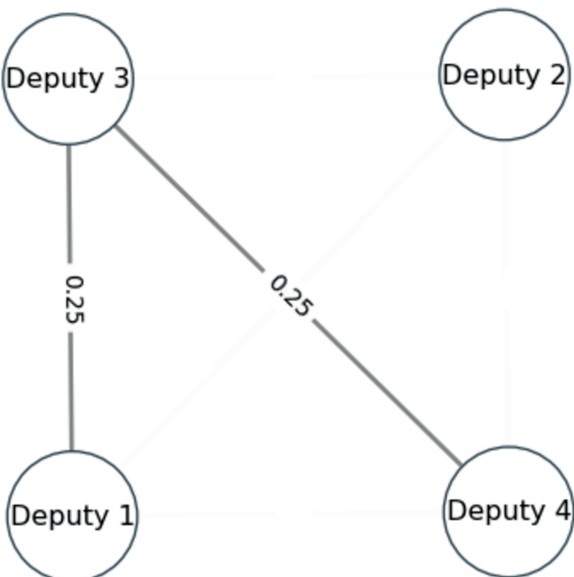

**Fig 3. Example voting network after keeping only positive weights.**

Since only edges with weight greater than zero are kept, the resulting network is shown in Fig 3.

## Backbone extraction

The backbone extraction is a methodology aiming to uncover the backbone of a network, which is a simpler representation of the network that identifies the patterns and communities easier [29]. Networks derived from human interactions are prone to generate noisy edges not representative of the phenomenon of interest [15]. Therefore, this methodology is useful.

The backbone extraction algorithm to be used in each network will be evaluated following two analyses: topological and contextual following [15].

The topological analysis will consist of verifying a set of resulting metrics from each backbone extraction algorithm in terms of two components: aggressiveness of pruning and quality of resulting communities. The contextual analysis will assess whether or not each of these algorithms can maintain the underlying phenomena intact. The idea of combining these two analyses is to uncover the smallest set of edges that reduce the noise of the network while preserving the representation of the phenomenon of interest [15], which is, in our case, the groupings of congress members.

To give more context on the quality of resulting communities from each backbone extraction, the following section discusses the quality metrics we chose to evaluate.

**Metric selection for backbone extraction evaluation.** Metrics to evaluate the quality of communities in a network can be controversial. Modularity is the metric most used, and the Surprise metric is a strong contender however, it suffers from the resolution problem in which smaller communities are usually undetected [30] and present distinct solutions of high score [31]. Both of these problems are reduced in the Surprise metric [32]. However, Surprise tends to overestimate the number of communities in a network [33]. Therefore, we consider both of these limitations in our pipeline and evaluate both metrics.

Modularity metric is defined as shown in Eq 1, where $k_i$ is the degree of node $i$, $m$ is the number of edges in the network, $A$ is the adjacency matrix of the network and $\delta$ is a function that yields 1 if $i, j$ are in the same community and zero otherwise.

$$Q = \frac{1}{2m} \sum_{ij} \left( A_{ij} - \frac{k_i k_j}{2m} \right) \delta(c_i, c_j) \tag{1}$$

The Surprise is defined as shown in Eq 2, where $n$ is the number of edges, $F$ is the maximum number of network edges, $M$ is the maximum number of edges inside the communities given a partition. The surprise measures how surprising it is to find a community with up to $l$ edges inside itself in a given graph.

$$S = -\log \sum_{j=l}^{\min(M,n)} \frac{\binom{M}{j}\binom{F-M}{n-j}}{\binom{F}{n}} \tag{2}$$

Selecting the backbone extraction methodology based only on the resulting modularity of the communities detected in the backbone network is misleading since there is a negatively correlated relationship between the network number of edges and its modularity, which happens because the modularity metric depends on the graph as whole and not only on the local topological characteristics of the communities. This can be seen in Fig 4 where the resulting modularity for the Senate, over time, for network versions where edges were randomly removed until the network acquired a desired density.

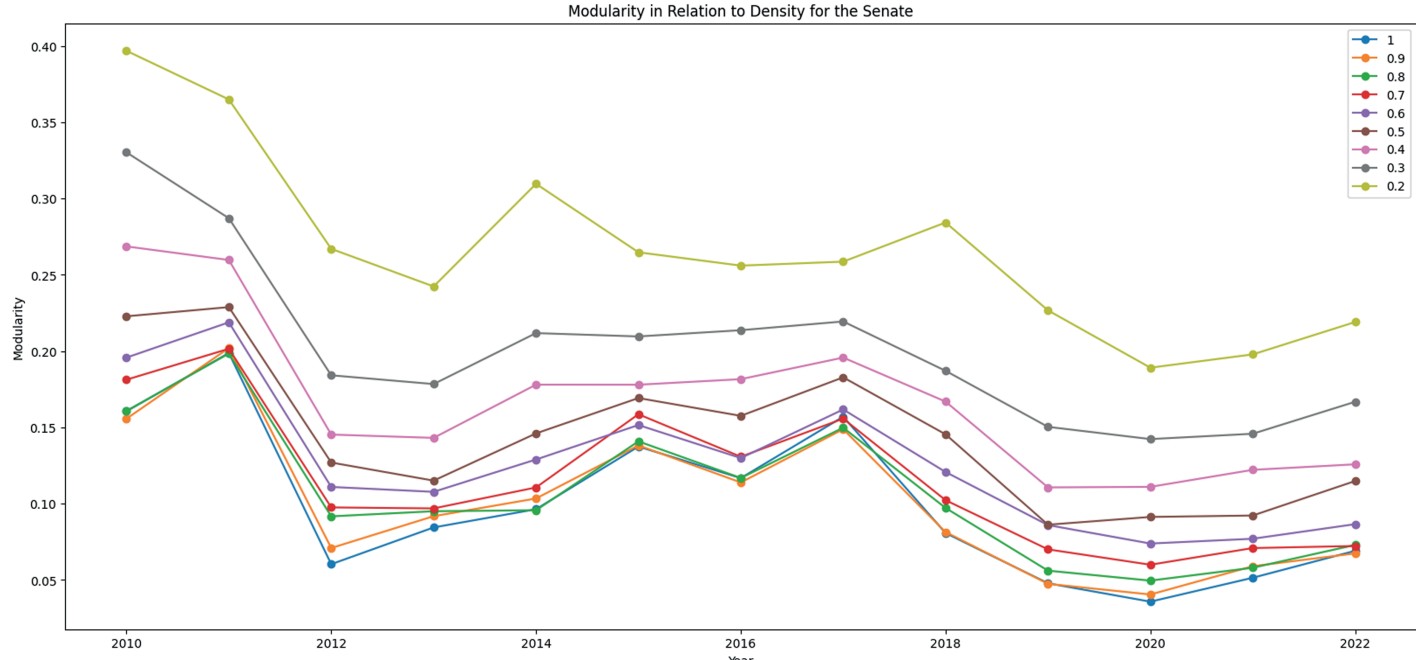

**Fig 4. Modularity x Density of the networks for randomly removed edges.**

Therefore, if we are comparing two algorithms considering their modularity values, then their overall density should be matched to avoid introducing a bias in comparison derived from the edge removal aggressiveness of the method.

On the other hand, the Surprise method, when applied to networks with a small set of communities, which is the case of legislative networks [1,11,17], tends to overestimate the number of communities [33].

Therefore, simply looking at the modularity values of two algorithms of different densities is misleading, optimizing modularity blindly can also introduce bias, but using a procedure that maximizes surprise can also introduce bias.

So, for our methodology, we apply the Leiden [19] algorithm, which maximizes modularity, to a varying grid of densities to allow for a fair comparison, but choose the method and density that yields the higher Surprise value, since it presents a nicer set of properties for our analysis. This way, we decide to balance the bias introduced by both algorithms.

**Selected methods.** In this work, the following methods were selected to be tested as the backbone extraction strategy:

- **Disparity Filter** - As proposed in [18].
- **Locally Adaptive Network Sparsification Filter (LANS)** - As proposed in [24].
- **High Salience Skeleton Filter** - As proposed in [25].
- **Doubly Stochastic Filter** - As proposed in [27].
- **Noise Corrected Filter** - As proposed in [26].

These methods were selected because they all allow filtering the remaining network to keep a desired fraction of the edges untouched.

**Topological analysis.** Following the pipeline description from [15], a topological analysis consisting of analyzing the metrics of the resulting networks is conducted.

The following metrics were selected to be evaluated in the pipeline in addition to modularity and Surprise:

- **Node Fraction** - Indicates the percentual value of remaining nodes in the network. This metric is evaluated to avoid methodologies that isolate too many nodes from the network.
- **LCC Size** - The number of nodes in the largest connected component. We aim to avoid highly fragmented networks, therefore we prefer methods in which most nodes are connected.

**Contextual analysis.** The second step of the analysis is called contextual analysis. Its objective is to verify if the resulting network, after the backbone extraction, still represents the underlying phenomenon that defines the edge weights between the nodes of the network. [15]. The steps to conduct the contextual analysis are as follows:

- Run every backbone extraction method. Here, we compare only the methods with the same fraction applied.
- Gather all the common edges for all algorithms and define it as a test set.
- For each algorithm, grab the set of obtained edges, that is disjointed from the test set and use it as a train set.
- Grab metadata information about the nodes to be used as features for training a predictive model.
- Train a machine learning model, for each backbone extraction algorithm, that predicts the edge weights of the resulting network.

- Assess the model performance in predicting the test set.

If the algorithm can achieve good predictive accuracy in the test set, in other words, is capable of identifying the edge weights from metadata, we say that the backbone extraction methodology did not remove the underlying phenomena. This final step is to guarantee that looking only at the topological metrics does not destroy the information we are trying to analyze by just optimizing a statistical metric.

An XGBoost Regressor [34] was selected as the model to predict the edge weights. We measured the resulting Mean Squared Error (MSE) of the prediction on the test set and used the following metadata as the features for prediction:

- Party of Node 1
- Party of Node 2
- Prevalence of Party from Node 1 in the respective house
- Prevalence of Party from Node 2 in the respective house

**Selected algorithms.** After both analyses have been conducted, the selection methodology is defined as follows, for each year and house of interest:

- Filter out every method in which the LCC Size is smaller than 80% of the nodes of the network;
- Filter out every method that could not be trained because of the non-overlapping test method;
- Select the method that yields the higher Surprise with the Leiden algorithm, removing ties by selecting the smaller MSE value.

In this way, the resulting methods are applied to every possible fraction, and all of them can predict the underlying effect since their MSE in the test set can be calculated.

## Community detection

After the backbone extraction step, the resulting networks are grouped into communities using a community detection algorithm. This step is necessary to investigate how groups behave in the legislative houses.

The selected community detection algorithm adopted for this task was the Leiden algorithm [19].

## Network characterization

All the groups' metrics have been defined according to those proposed in [1]. A group can be a party or a detected community. The coalition of two groups is computed as

$$d(A,B) = \frac{1}{|A \times B|} \sum_{(a,b) \in A \times B} l(a,b), \tag{3}$$

where $l(A,B)$ is the shortest path length between two nodes, one from each group $A$ and $B$.

This metric represents the average spatial separation between elements belonging to different groups. When members from both groups consistently vote in alignment, their distance is expected to be minimal, indicating a stronger coalition between the two groups.

Since $l(A, B)$ cannot be directly calculated on the generated network since edges present a similarity index between nodes, the edge weights were transformed into a dissimilarity index according to Eq 4, following the approach from [1].

$$\Delta\left(w_{ij}\right) = 2\left(1 - w_{ij}\right)^{\frac{1}{2}} \tag{4}$$

The isolation of a group $A$ is computed as

$$I(A) = \frac{\sum_{X \neq A} |X| d(A, X)}{\sum_{X \neq A} |X|} \tag{5}$$

This metric represents the average distance between $A$ and all other groups within the network. As a group becomes increasingly isolated from others, its average distance is expected to be larger.

The fragmentation of a group can be computed as $F(A) = d(A, A)$. This metric assesses the level of intra-group dispersion. When nodes representing members of the same group are situated far apart from each other in the network, it indicates reduced cohesiveness within the group.

To measure the effective number of groups, parties, or communities, in the networks, the concept of true diversity [35] is applied. This measurement considers each group's size when calculating the effective number of groups.

Let $\Pi = \{\pi_1, \pi_2, \ldots, \pi_k\}$ be the set of the group normalized sizes being analyzed. The true diversity of this set is computed as

$$D = \exp\left(-\sum_{i=1}^{k} \pi_i \ln \pi_i\right) \tag{6}$$

This concept has been widely employed in various contexts to quantify the effective number of items based on the analysis of the size distribution across different classes [36,37].

**Government and opposition.** To define which parties are members of the government and the opposition, the coalition of the government party and all other parties in the network is calculated.

Then, the parties with a coalition smaller than the 15% percentile of the coalition distribution for our data are defined as the government parties. The parties with a coalition greater than the 85% percentile of the coalition distribution are set as the opposition parties. All parties in between are defined as neutral.

Formal coalitions in Brazil are usually not represented in terms of voting behavior since it is common to see betrayals [11] and continuous effort from the government to please members of its base. Therefore, we chose a methodology that is gathered entirely from data and represents the actual action of parties (not their promises).

**Communities analysis.** The resulting communities are analyzed given their internal distribution of parties, political spectrum, and government/opposition parties.

We expected that by analyzing these distributions we will find out political alliances, both in terms of parties and political spectrum, related to the government party. We also expect to find parties that behave against their political spectrum.

## Dataset and network assessment

### Data source

All the data for the Brazilian Chamber of Deputies was extracted via the csv files and the REST API of the project "Dados Abertos da Câmara" (Chamber Open Data - Available in https://dadosabertos.camara.leg.br/). For the Senate data, the system "Dados Abertos do Senado" (Senate Open Data - Available in https://www12.senado.leg.br/dados-abertos) was used.

### Legislative houses sizes

In Fig 5, the number of propositions, by year, for the Senate and the Chamber of Deputies is shown.

The Chamber of Deputies presents a higher number of non-anonymous propositions being voted on each year in comparison with the Senate. However, the number of propositions in both houses is highly correlated, having a Pearson correlation of 0.79.

In Fig 6, it can be seen the cumulative sum of the number of deputies over time by political party. The same is found for the Senate in Fig 7. It is important to notice that fluctuation in the number of deputies and senators occurs annually due to the inclusion of substitute members taking up their positions during their mandates.

The results reveal that the Senate is mainly comprised of 9 parties, accounting for over 65% of the total number of senators. However, these 9 parties do not dominate the Chamber of Deputies as much since they constitute only about half of the house. The presence of smaller parties (called 'Others' in the graph) has grown significantly in the last few years, whereas it has been approximately constant in the Senate.

Therefore, there are significant structural differences between both houses, at least in terms of the number of propositions voted and the party structure dominance.

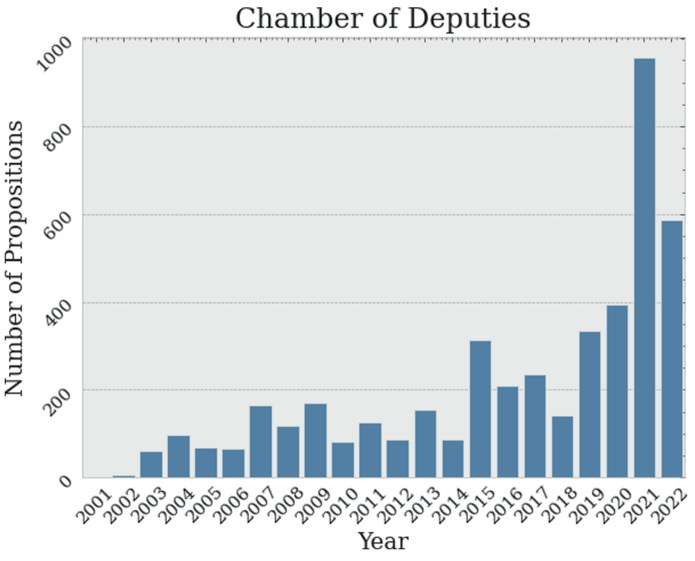
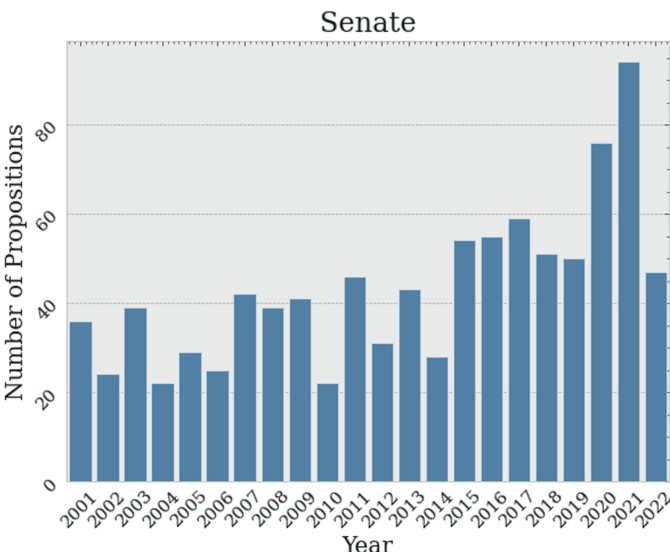

**Fig 5. Number of yearly propositions for the Senate and the Chamber of Deputies.**

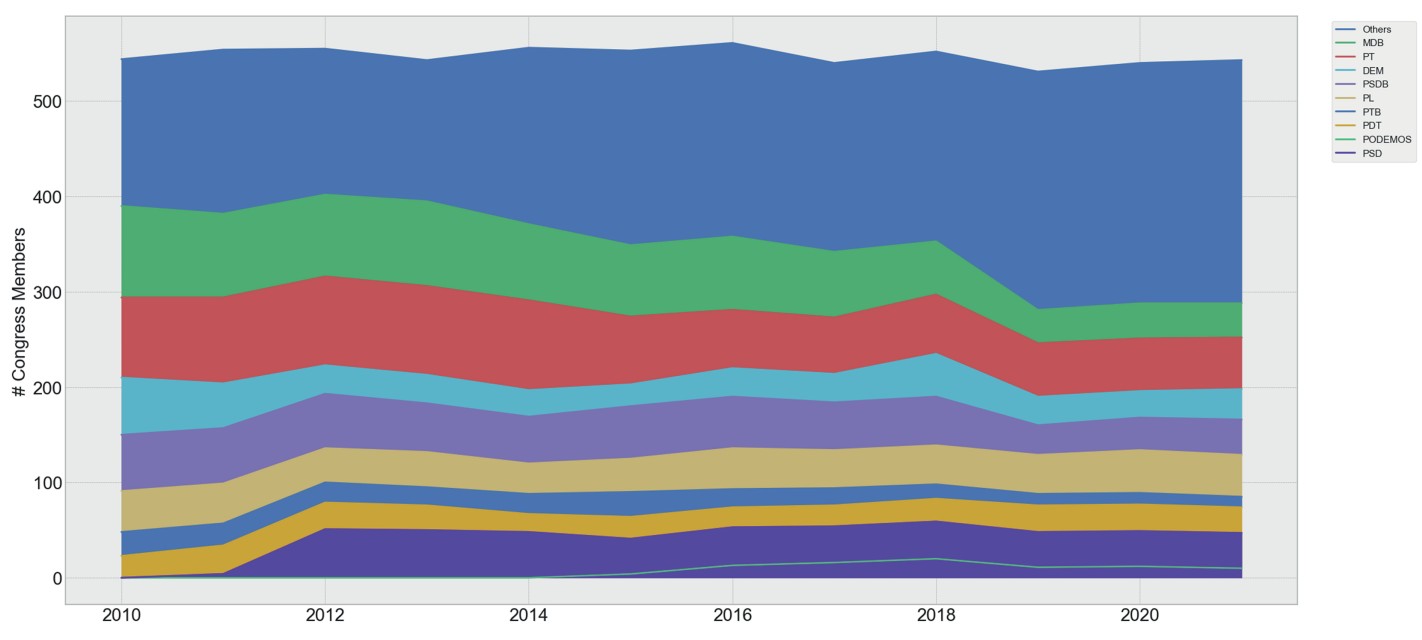

**Fig 6. Brazilian Deputies Chamber cumulative members distribution.**

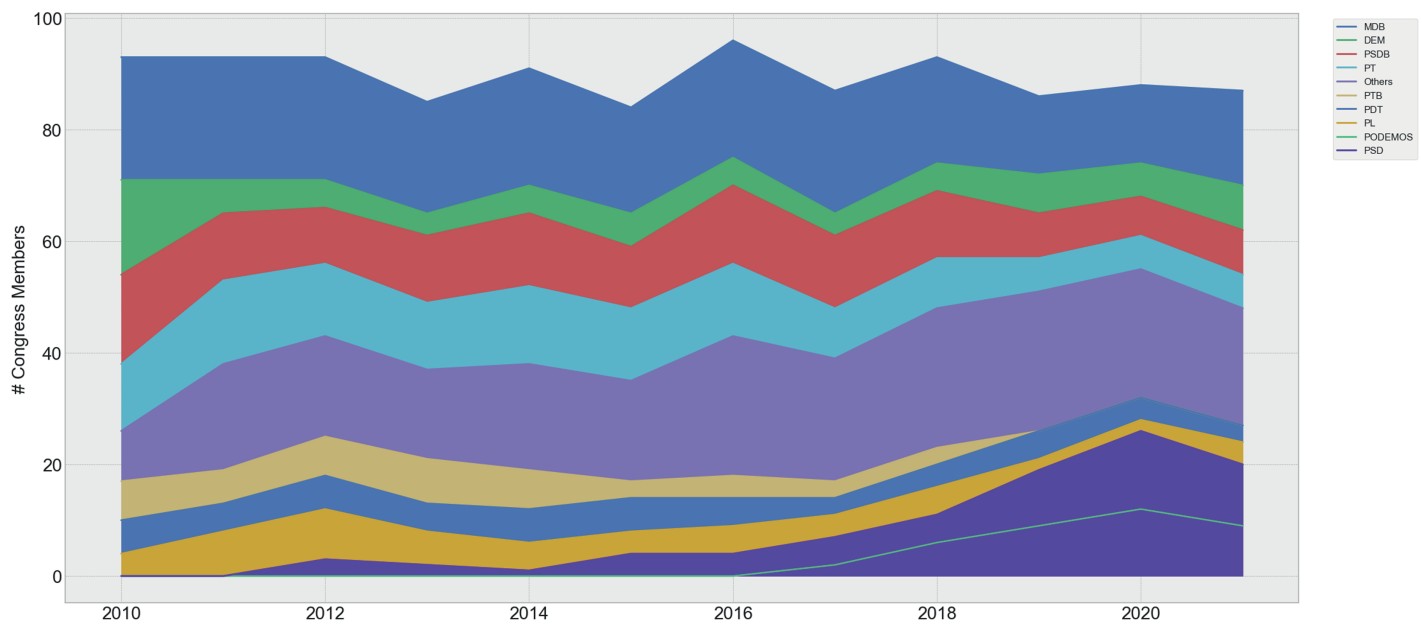

**Fig 7. Senate cumulative members distribution.**

Also, it is shown that the average number of members of the Chamber of Deputies stays around 500 year-over-year, while for the Senate there are around 90 senators.

Finally, the total number of parties for each year, for both, the Senate and the Chamber of Deputies, can be seen in Fig 8.

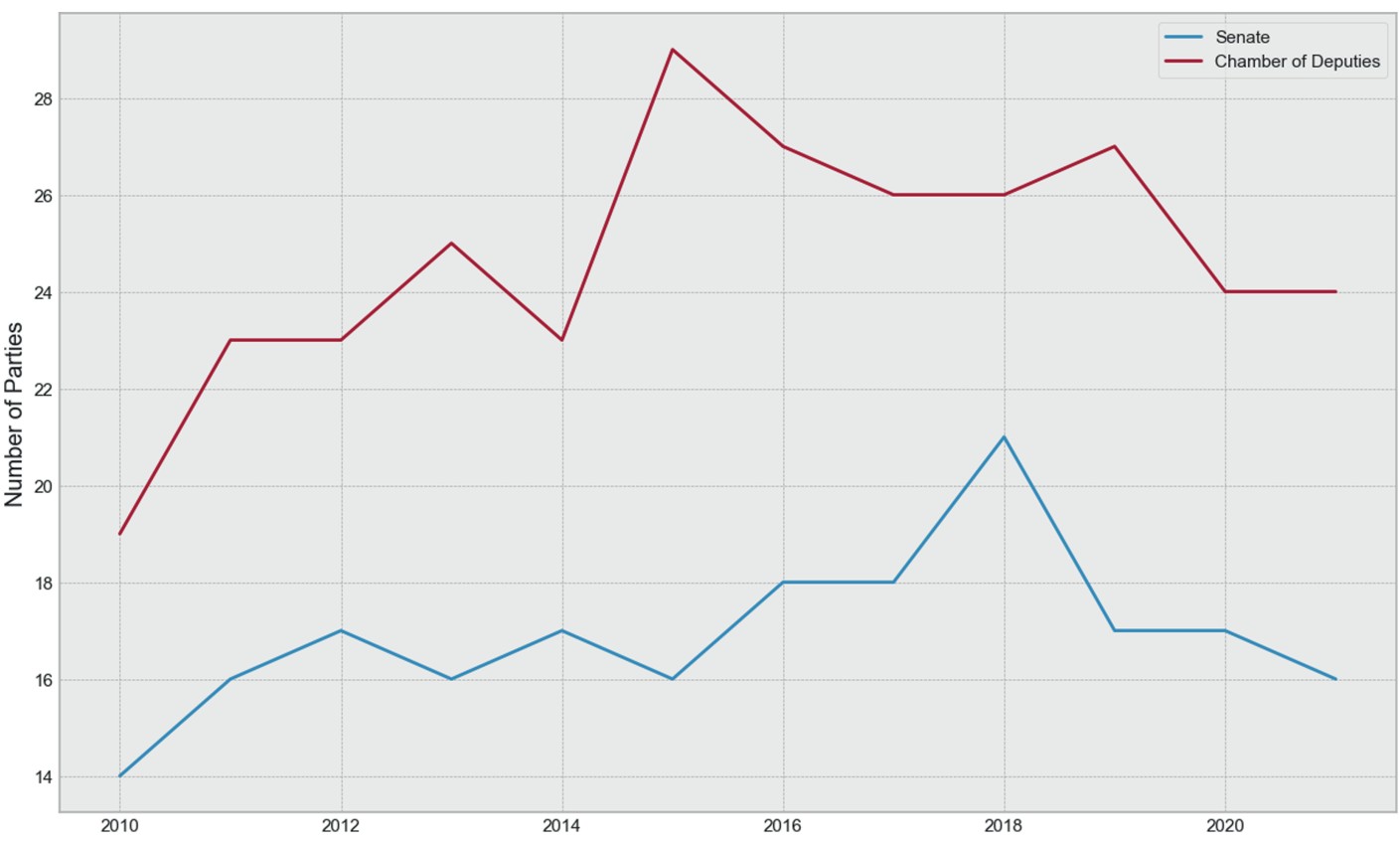

**Fig 8. Number of parties over time for both houses.**

## Parties and political spectrum

To adjust for party name changes, we consider only the most recent name, until 2021 and rename old instances accordingly. In Table available in Annex 1 is shown the final party name, the old names considered, and the political spectrum of the respective party. The spectrum for each party was gathered from their official website in July 2023.

## Network statistics

The statistics of Brazilian Deputies Chamber network, by year, are presented in Table 1 while in Table 2 are the statistics by year for the Senate networks, where N denotes the number of nodes, M is the number of edges, Components is the number of connected components, Clustering C is the average clustering coefficient and Degree is the average degree. Both are generated before the backbone extraction methodology.

## Results and discussion

### Backbone extraction

To verify if the best backbone extraction algorithm is the same for both houses, we will analyze the resulting metrics and algorithm choices done by the proposed pipeline.

**Table 1. Network statistics for Brazilian Deputies Chamber before backbone extraction application.**

| Year | N | M | Components | Clustering C. | Transitivity | Assortativity | Degree |
|---|---|---|---|---|---|---|---|
| 2010 | 544 | 129634 | 2 | 0.21 | 0.92 | 0.00 | 476.5 |
| 2011 | 554 | 107439 | 2 | 0.31 | 0.95 | 0.24 | 387.8 |
| 2012 | 555 | 128872 | 2 | 0.17 | 0.90 | 0.02 | 464.4 |
| 2013 | 543 | 128415 | 2 | 0.18 | 0.92 | 0.01 | 472.9 |
| 2014 | 556 | 148781 | 2 | 0.23 | 0.97 | -0.03 | 535.1 |
| 2015 | 553 | 120084 | 2 | 0.17 | 0.85 | 0.00 | 434.3 |
| 2016 | 561 | 109447 | 2 | 0.27 | 0.92 | 0.16 | 390.1 |
| 2017 | 540 | 96836 | 2 | 0.24 | 0.93 | 0.20 | 358.6 |
| 2018 | 552 | 98222 | 2 | 0.24 | 0.92 | 0.20 | 355.8 |
| 2019 | 531 | 95684 | 2 | 0.30 | 0.94 | 0.29 | 360.3 |
| 2020 | 540 | 98366 | 1 | 0.37 | 0.94 | 0.28 | 364.3 |
| 2021 | 543 | 97704 | 1 | 0.35 | 0.94 | 0.24 | 359.8 |

**Table 2. Network statistics for the Senate before backbone extraction application.**

| Year | N | M | Components | Clustering C. | Transitivity | Assortativity | Degree |
|---|---|---|---|---|---|---|---|
| 2010.0 | 93.0 | 1135 | 2 | 0.11 | 0.60 | 0.13 | 24.4 |
| 2011.0 | 93.0 | 1550 | 3 | 0.12 | 0.71 | -0.00 | 33.3 |
| 2012.0 | 93.0 | 2313 | 2 | 0.16 | 0.78 | -0.06 | 49.7 |
| 2013.0 | 85.0 | 2040 | 6 | 0.17 | 0.84 | -0.15 | 48.0 |
| 2014.0 | 91.0 | 1706 | 3 | 0.13 | 0.74 | -0.04 | 37.4 |
| 2015.0 | 84.0 | 2023 | 4 | 0.18 | 0.83 | -0.14 | 48.1 |
| 2016.0 | 96.0 | 2485 | 3 | 0.14 | 0.78 | -0.10 | 51.7 |
| 2017.0 | 87.0 | 2283 | 2 | 0.18 | 0.82 | -0.12 | 52.4 |
| 2018.0 | 93.0 | 1927 | 3 | 0.13 | 0.73 | -0.14 | 41.4 |
| 2019.0 | 86.0 | 2640 | 3 | 0.26 | 0.93 | -0.01 | 61.3 |
| 2020.0 | 88.0 | 3508 | 2 | 0.40 | 0.96 | -0.08 | 79.7 |
| 2021.0 | 87.0 | 3138 | 3 | 0.38 | 0.95 | -0.06 | 72.1 |

The resulting metrics for the analysis can be found in Annex 2. The selected fraction and the selected backbone extraction methodology for each year can be found in Table 3 for the Chamber of Deputies and in Table 4 for the Senate.

The Doubly Stochastic Filter yields, on average, the greatest modularity among all methods, in terms of Surprise, for smaller networks. It consistently overcomes other methods and is competitive for bigger fractions until around 2017. It also has a similar LCC size to other methods. This justifies its selection for most of the Senate and for the starting years of the Chamber of Deputies. It has not lost performance in recent years for the Senate.

For more recent networks in Brazilian Deputies Chamber, their MSE appears less competitive. Still, an overall increase in the MSE is found for all methods, which indicates that the group dynamics of the networks may have changed from 2018 onward, which comprises the last year of the government from President Michel Temer and the beginning of the government for President Jair Bolsonaro.

Furthermore, it is relevant to notice that for older networks, smaller fractions are selected, while bigger fractions appear more frequently after 2016, which can also indicate a change in the dynamics of the houses.

The resulting network statistics, after application of the backbone extraction methodology, are shown, for the Chamber of Deputies, in Table 5, and for the Senate in Table 6.

**Table 3. Resulting backbone extraction methods and fractions for the Chamber of Deputies.**

| Year | Method | Fraction of Edges |
|---|---|---|
| 2010 | Doubly Stochastic Filter | 0.2 |
| 2011 | High Salience Skeleton Filter | 0.9 |
| 2012 | Doubly Stochastic Filter | 0.3 |
| 2013 | Doubly Stochastic Filter | 0.3 |
| 2014 | Doubly Stochastic Filter | 0.3 |
| 2015 | Doubly Stochastic Filter | 0.4 |
| 2016 | High Salience Skeleton Filter | 0.8 |
| 2017 | Doubly Stochastic Filter | 0.7 |
| 2018 | Noise Corrected | 0.7 |
| 2019 | High Salience Skeleton Filter | 0.8 |
| 2020 | Noise Corrected | 0.8 |
| 2021 | High Salience Skeleton Filter | 0.8 |

**Table 4. Resulting backbone extraction methods and fractions for the Senate.**

| Year | Method | Fraction of Edges |
|---|---|---|
| 2010 | Doubly Stochastic Filter | 0.5 |
| 2011 | Doubly Stochastic Filter | 0.5 |
| 2012 | Doubly Stochastic Filter | 0.5 |
| 2013 | LANS | 0.3 |
| 2014 | Doubly Stochastic Filter | 0.3 |
| 2015 | Doubly Stochastic Filter | 0.7 |
| 2016 | Doubly Stochastic Filter | 0.4 |
| 2017 | Doubly Stochastic Filter | 0.5 |
| 2018 | Doubly Stochastic Filter | 0.5 |
| 2019 | Doubly Stochastic Filter | 0.3 |
| 2020 | Doubly Stochastic Filter | 0.3 |
| 2021 | Doubly Stochastic Filter | 0.2 |

**Table 5. Network statistics for Brazilian Deputies Chamber after the backbone extraction application.**

| Year | N | M | Components | Clustering C. | Transitivity | Assortativity | Degree |
|---|---|---|---|---|---|---|---|
| 2010 | 541 | 25790 | 1 | 0.19 | 0.63 | 0.23 | 95.34 |
| 2011 | 553 | 96696 | 1 | 0.35 | 0.95 | 0.22 | 349.71 |
| 2012 | 553 | 37812 | 1 | 0.18 | 0.64 | 0.20 | 136.75 |
| 2013 | 542 | 37255 | 1 | 0.17 | 0.65 | 0.20 | 137.47 |
| 2014 | 555 | 44275 | 1 | 0.21 | 0.69 | 0.09 | 159.55 |
| 2015 | 552 | 41392 | 1 | 0.22 | 0.71 | 0.12 | 149.97 |
| 2016 | 560 | 87558 | 1 | 0.33 | 0.93 | 0.19 | 312.71 |
| 2017 | 538 | 52717 | 1 | 0.25 | 0.86 | 0.48 | 195.97 |
| 2018 | 527 | 68756 | 1 | 0.34 | 0.94 | 0.22 | 260.93 |
| 2019 | 530 | 76548 | 1 | 0.36 | 0.94 | 0.35 | 288.86 |
| 2020 | 523 | 78693 | 1 | 0.45 | 0.95 | 0.37 | 300.93 |
| 2021 | 543 | 78164 | 1 | 0.42 | 0.94 | 0.26 | 287.90 |

The methodology chosen for each year kept most of the original nodes while generating a single connected component.

**Table 6. Network statistics for the Senate after the backbone extraction application.**

| Year | N | M | Components | Clustering C. | Transitivity | Assortativity | Degree |
|------|-----|-----|-----------|---------------|--------------|---------------|--------|
| 2010 | 83 | 288 | 1 | 0.08 | 0.32 | 0.51 | 6.94 |
| 2011 | 79 | 272 | 1 | 0.10 | 0.35 | 0.29 | 6.89 |
| 2012 | 84 | 426 | 1 | 0.12 | 0.42 | 0.33 | 10.14 |
| 2013 | 80 | 408 | 1 | 0.23 | 0.35 | -0.16 | 10.20 |
| 2014 | 80 | 282 | 1 | 0.10 | 0.40 | 0.57 | 7.05 |
| 2015 | 76 | 405 | 1 | 0.25 | 0.37 | -0.29 | 10.66 |
| 2016 | 89 | 394 | 1 | 0.14 | 0.48 | 0.46 | 8.85 |
| 2017 | 79 | 372 | 1 | 0.16 | 0.49 | 0.31 | 9.42 |
| 2018 | 79 | 316 | 1 | 0.10 | 0.36 | 0.39 | 8.00 |
| 2019 | 78 | 261 | 1 | 0.20 | 0.48 | 0.24 | 6.69 |
| 2020 | 86 | 701 | 1 | 0.33 | 0.66 | 0.10 | 16.30 |
| 2021 | 83 | 621 | 1 | 0.31 | 0.64 | 0.14 | 14.96 |

Besides that, as it was expected, the number of edges is greatly reduced as well as the degree and the Clustering Coefficient for both houses changed. However, the values are kept around the same magnitude as the original networks.

It is important to highlight that the methods reduced the transitivity of the network, but improved the assortativity of degree, indicating that nodes are now more connected to nodes with similar degrees.

**Network visualization.** In Fig 9, the networks generated for the year 2013, for both houses, before and after the backbone extraction methodology are shown.

It can be seen that the backbone extraction methodology uncovered a more prominent community structure and it was able to group up political parties together, a dynamic that was not clear on the raw network.

Finally, it is shown that the best algorithm for each year and network is different based on the modularity and the MSE metric of the resulting networks. This shows that, for each year, the algorithm that yields the higher value for the metrics we are evaluating, there is no *one size fits all* algorithm for the temporal variation of the networks. A more detailed view of those metrics for each year, on which the conclusion was based, is found in the Supporting Information. Therefore, using the same algorithm on both houses without testing for its performance can be misleading. This is an improvement over the methodology applied in [1,11].

## Communities and political parties

Now, to verify if the community structure is similar for both houses, it is necessary to analyze the found communities in terms of parties, political spectrum, and government alignment.

The number of communities found for Brazilian Deputies Chamber and the Senate, by year, is shown in Fig 10. One can see that the number of communities found is smaller than the number of parties for both houses. This finding was also shown extensively for Brazilian Deputies Chamber and other countries in previous research [1,11,13–15,23], but is shown for the first time for the Brazilian Senate.

In Fig 11, the modularity for the parties and communities, for both the Senate and the Chamber of Deputies, are shown. The presented modularity for the communities is consistently higher than the one presented by the parties, which is in line with the findings from [1]. However, the presented modularity here is smaller than the previous study because of the

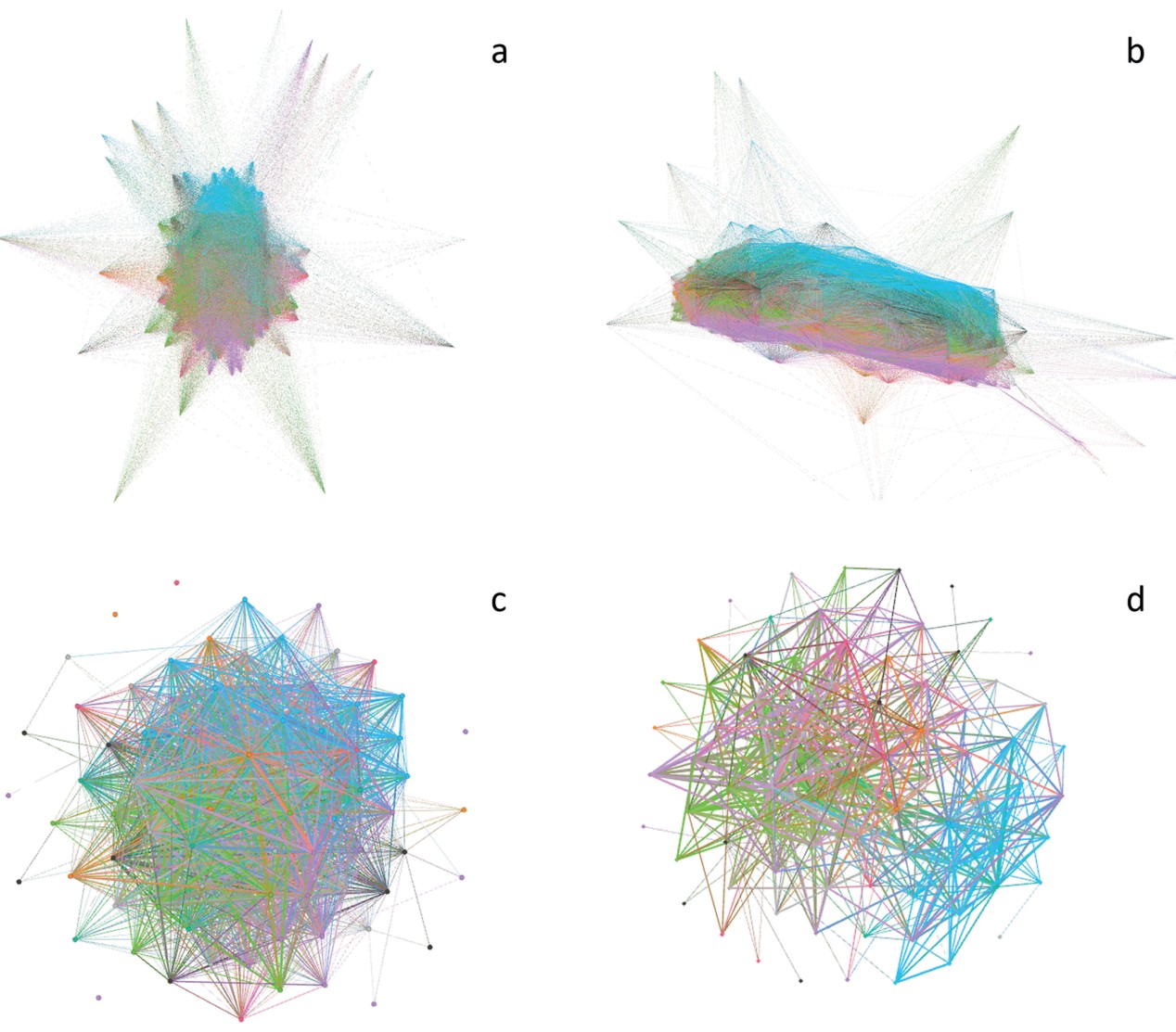

**Fig 9. The resulting networks with colors representing political parties.** a) The raw Brazilian Deputies Chamber network for 2013. b) Brazilian Deputies Chamber network after the backbone extraction. c) The raw Senate network for 2013. d) The Senate network after the backbone extraction. Visualizations created with Gephi.

process in which Surprise is maximized instead of the modularity to improve the quality and stability of the communities.

Finally, in Figs 12 and 13, it is shown that the Surprise is also consistently higher for the communities, as compared to the parties.

These two results show once again that the party structure is not the structure that maximizes modularity, and it does not maximize Surprise. Therefore, there is an underlying community structure that better represents the relationships between congress members. This is also valid for the Senate, as is shown for the first time.

This is also shown, by the resulting number of effective groups, for both houses in Fig 14. It is valid to highlight that the number of communities and effective communities is close to

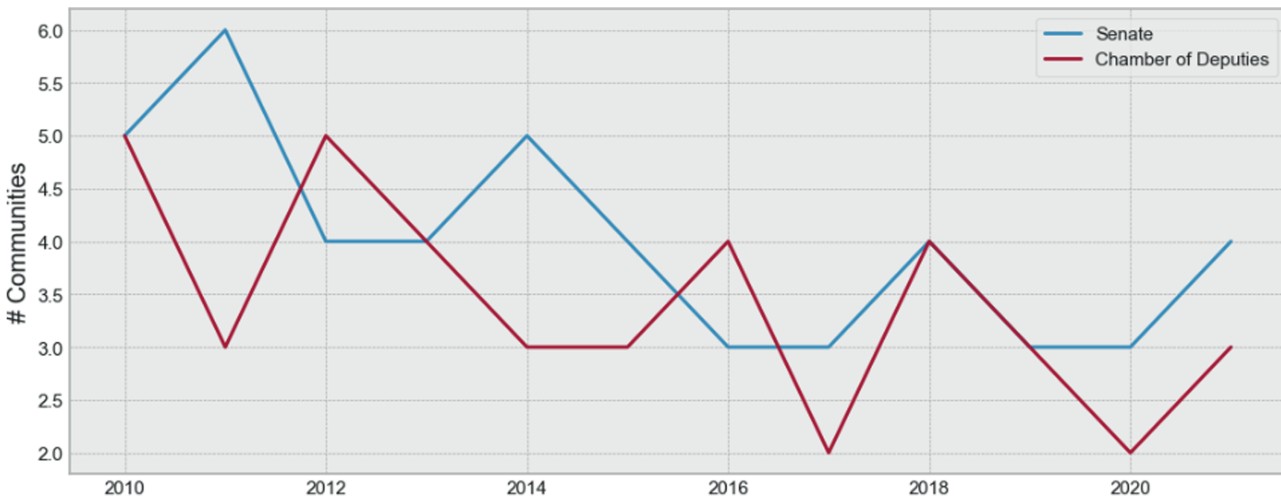

**Fig 10. A comparison of the number of found communities for the Senate and the Chamber of Deputies.**

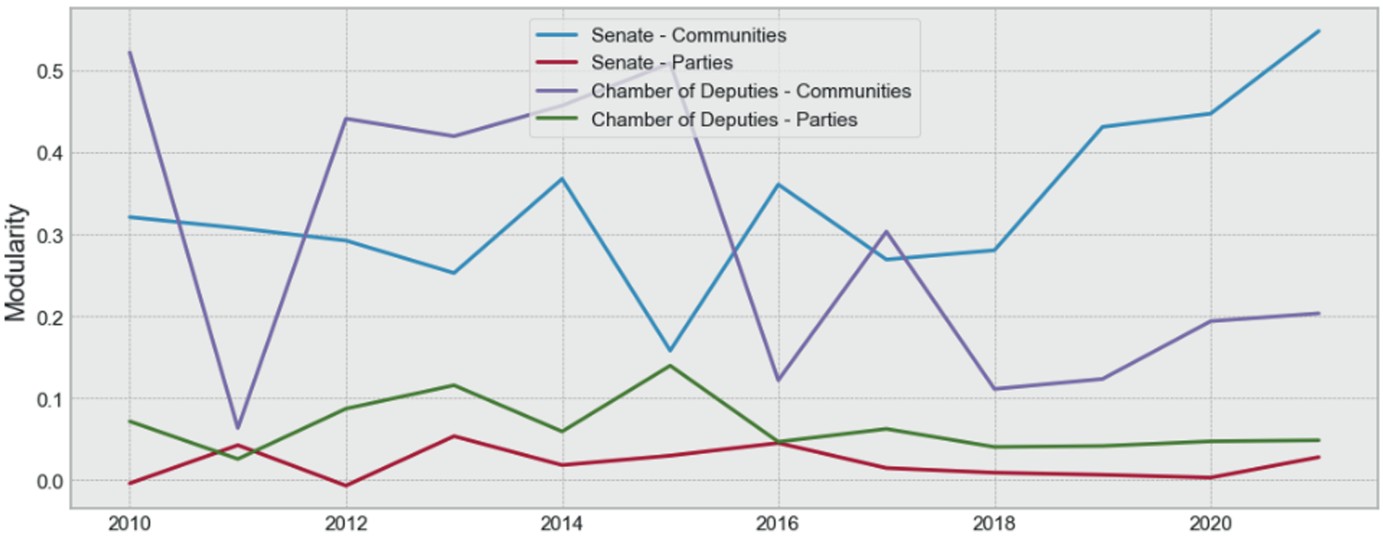

**Fig 11. Comparison of the modularity for parties and communities for the Senate and the Chamber of Deputies.**

one for both houses, showing that the communities are a good representation to analyze the behavior of congress members. This is also in line with the findings from [1].

**Party distribution in communities.** Some interesting findings were made by analyzing the treemap of the generated communities and visualizing which parties were associated with each group. First, during 2013, a year of great pressure over President Dilma Rousseff because of the popular manifestations, one would expect some kind of legislative difficulty for the governing party (PT), however, there is no signal of strong isolation in any of the houses. However, it appears that the party suffered from high fragmentation in the Chamber of Deputies, but not in the Senate. The Senate also appears to have a more cohesive opposition in this year. These findings are present in Fig 15.

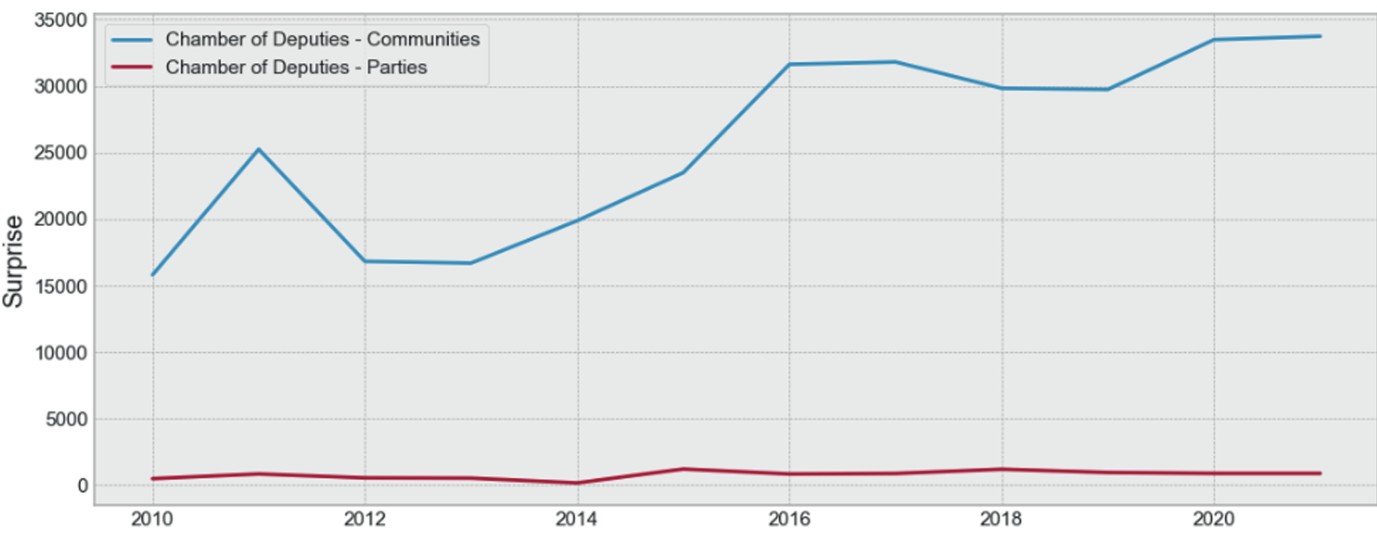

**Fig 12. Comparison of the Surprise for parties and communities for the Chamber of Deputies.**

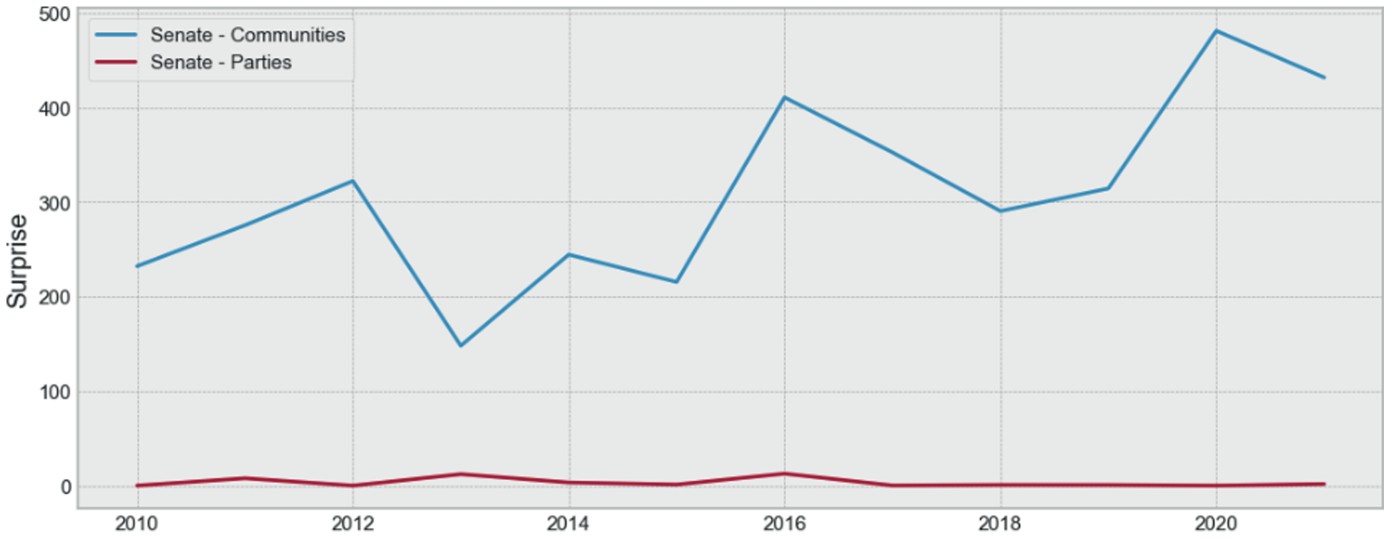

**Fig 13. Comparison of the Surprise for parties and communities for the Senate.**

In 2016 the leftist group, mostly represented by the PT, was isolated into a single group in the Chamber of Deputies. This year was the year of the Impeachment process of Dilma Rousseff. However, this process is only represented in the Senate for the 2017 network and even at that point, the isolation is less severe than the one faced in the Chamber of Deputies. The treemaps for this analysis are presented in Fig 16.

After that, the leftist parties, especially PT, continued isolated mostly in a single group, heavily associated with other leftist parties for both the Senate and the Chamber of Deputies.

**Political spectrum distribution in communities.** Following the isolation of PT during the years 2016 and 2017, the political spectrum of the communities also shows an isolation of the Left in a single group for the Chamber of Deputies. Once again, the Senate seems to be

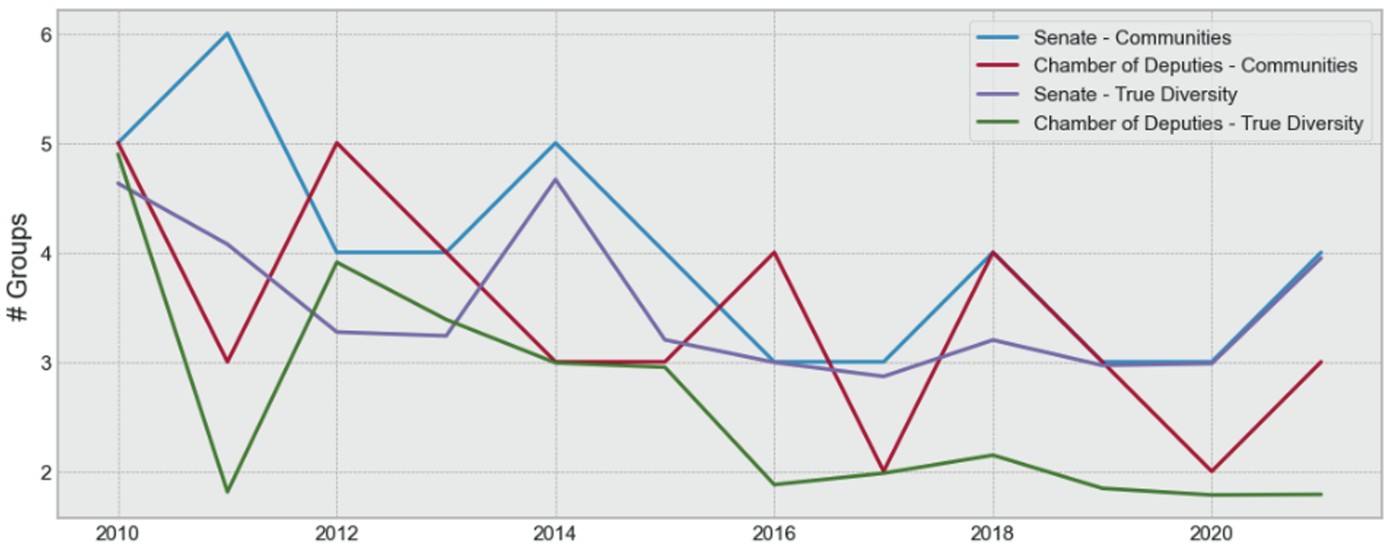

**Fig 14. Comparison between the number of communities and the true diversity of communities for the Senate and the Chamber of Deputies.**

Chamber: 2013

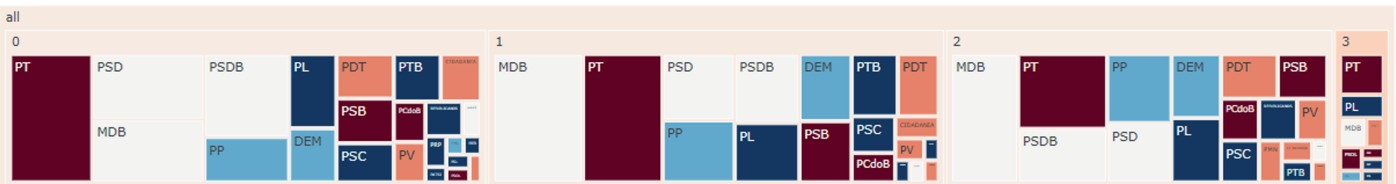

Senate: 2013

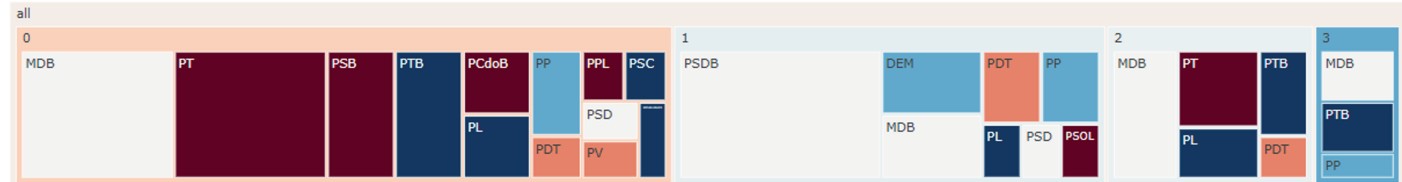

**Fig 15. The party distribution over communities for 2013.** Colors represent the political spectrum of parties with red being more to the left and blue more to the right.

less volatile to this kind of change, keeping communities with smaller segmentation. This can be found in Supporting Information 3.

**Government and opposition distribution in communities.** The year 2013 is the first year in which communities appear to be more focused on opposition, neutrals, and government for the Senate, but not for the Chamber of Deputies as shown in Fig 17.

This also happens again during 2016 but this time more focused in Brazilian Deputies Chamber, that appears to have a more isolated government community as shown in Fig 18.

Chamber: 2016

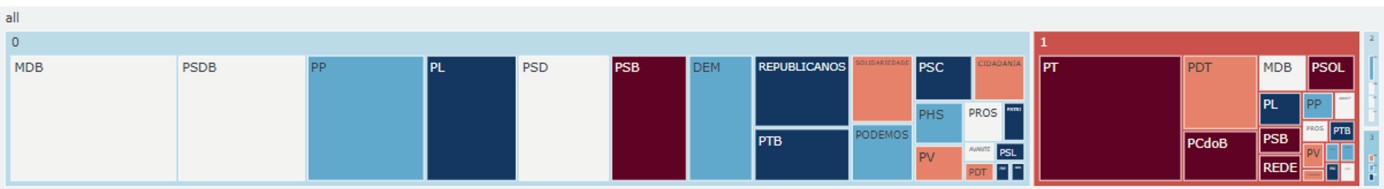

Senate: 2016

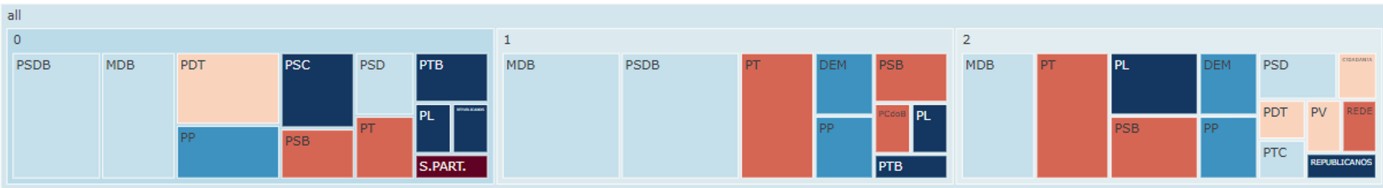

Chamber: 2017

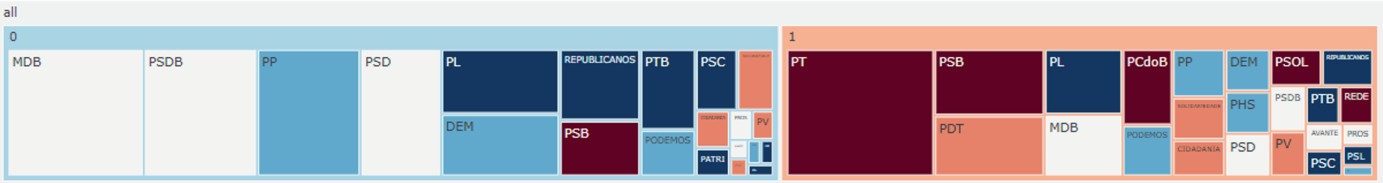

Senate: 2017

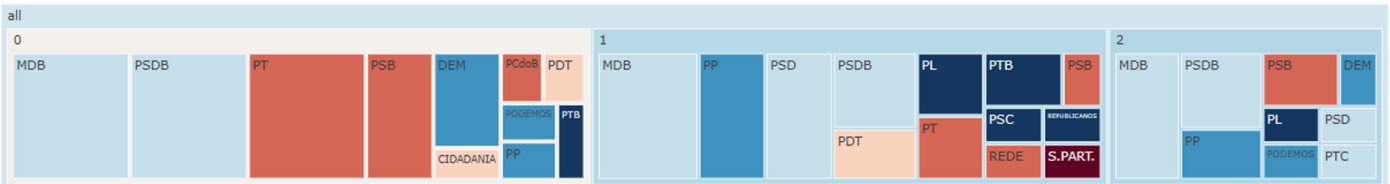

**Fig 16. The party distribution over communities for 2016 and 2017.** Colors represent the political spectrum of parties with red being more to the left and blue more to the right.

However, with a greater impact on the Chamber of Deputies. Once again, the Senate seems to be less volatile.

## Isolation and fragmentation analysis

Finally, to verify if the behavior is the same between houses over time for the political spectrum of the parties, the fragmentation and isolation analysis was conducted.

Each party was assigned a rank given their isolation and fragmentation. For example, the least isolated party for a given year received a rank of 1, while the most isolated received the highest possible rank value. Then, the average rank for each political spectrum was calculated for each year.

Fig 19 shows the fragmentation over time for the Chamber of Deputies. It can be seen that the Left becomes more fragmented in 2012 and 2013, years of high pressure over the leftist

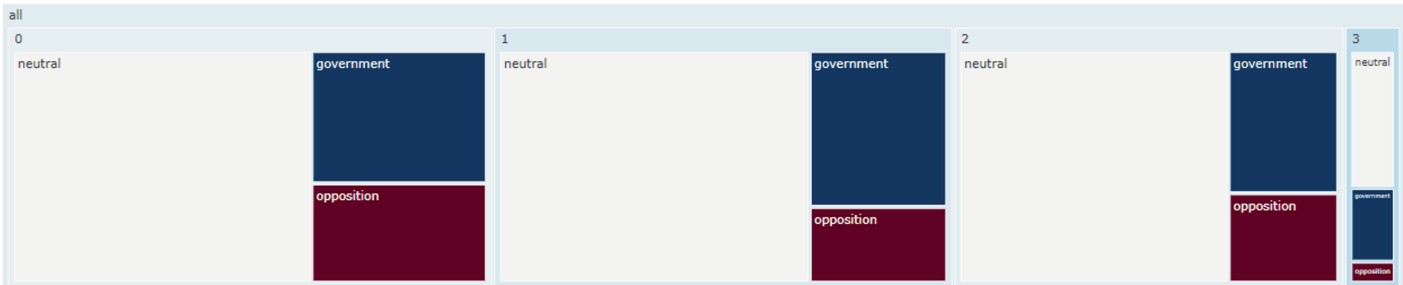

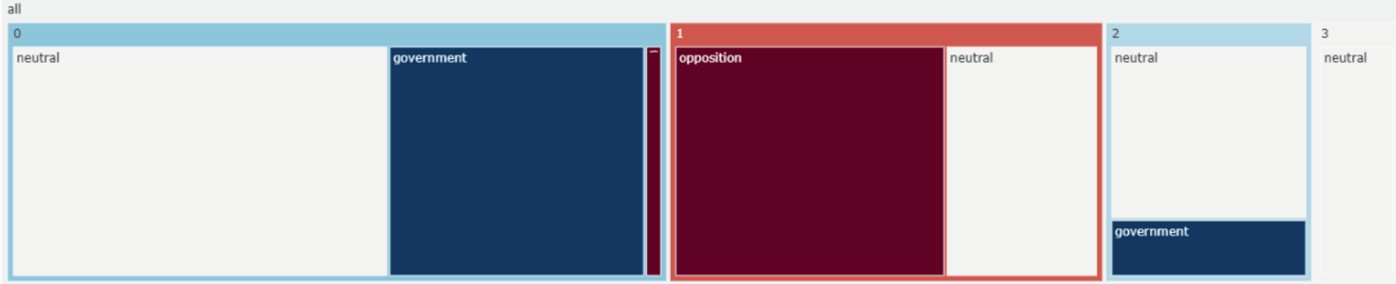

**Fig 17. The government x opposition distribution inside communities for 2013.**

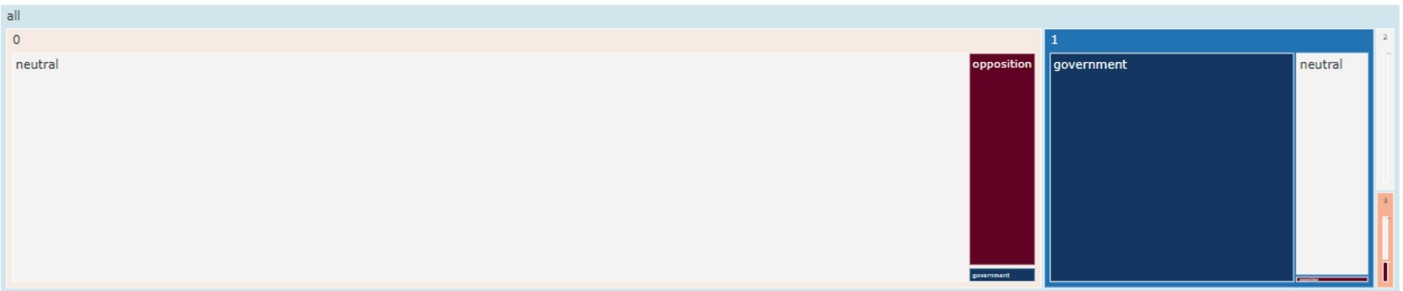

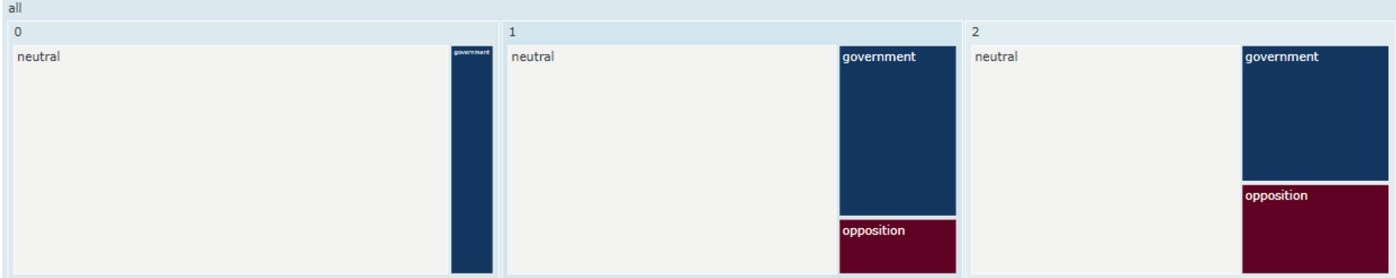

**Fig 18. The government x opposition distribution inside communities for 2016.**

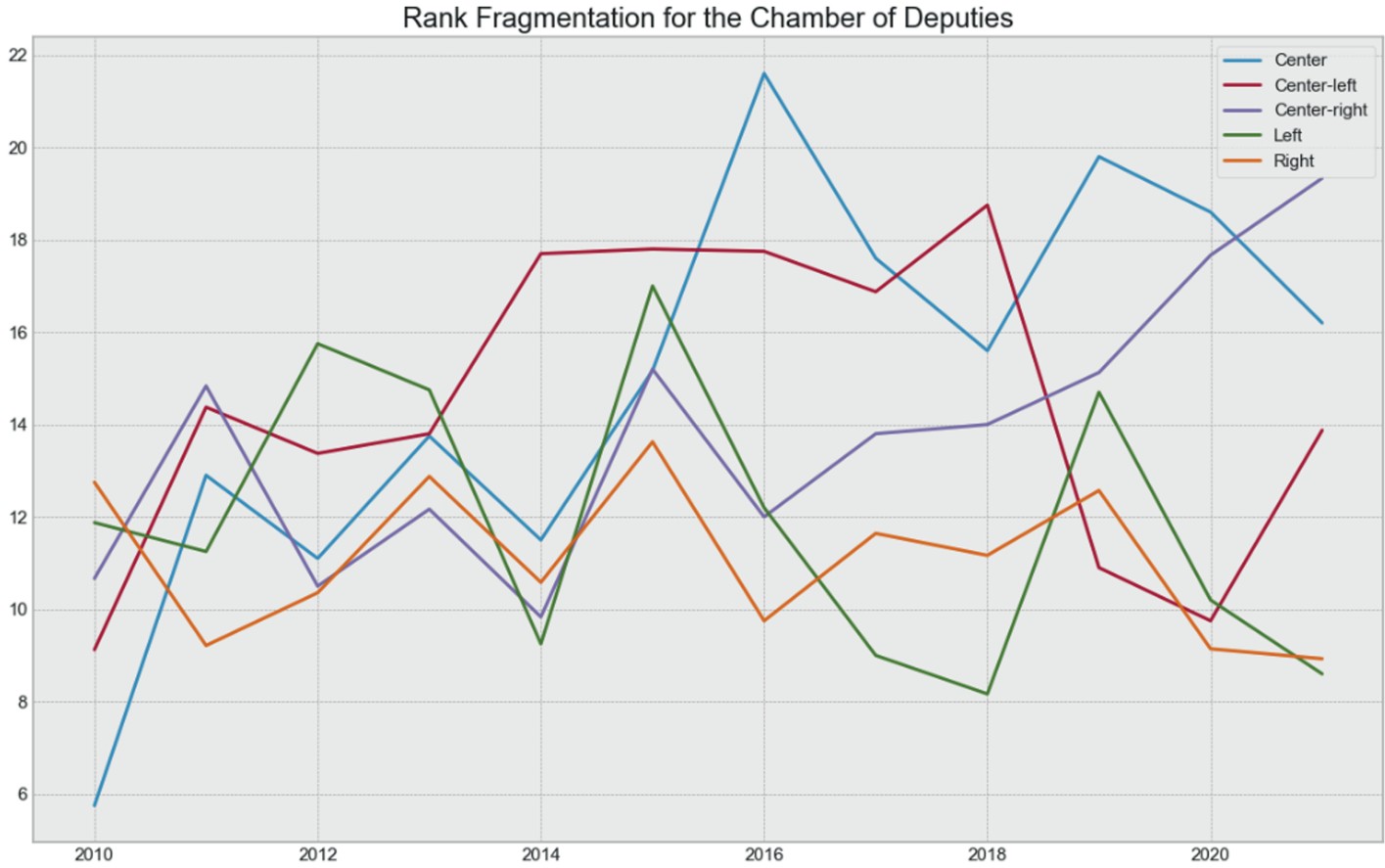

**Fig 19. Chamber of Deputies fragmentation rank by year.**

president. After that, they became one of the least fragmented spectrum, after the impeachment process and the election of the right-wing president in 2019. The fragmentation of the right stays low, during the entire period, but especially in the year of the Impeachment, and after the new mandate in 2019. The opposite is seen in the Senate in Fig 20, where the Left has low fragmentation during the pressure years, but becomes more fragmented after the impeachment. The low fragmentation of the Right after 2018 remains in the Senate.

Now, in terms of isolation, Fig 21 shows that the Left was mostly isolated in 2015 and 2016, the years related to the impeachment, and stayed as the most isolated spectrum until the end of the series. This phenomenon is not reflected in the Senate, as seen in Fig 22, which shows that the Center appears to be the most isolated spectrum for that house. On the other hand, the Right was the least isolated group during Jair Bolsonaro's mandate.

This shows that the Left suffered a bigger hit in the Chamber of Deputies, compared to the Senate during Dilma Rousseff's second mandate, and that the Right had a highly cohesive and non-isolated group for the first years of Jair Bolsonaro's mandate.

## Chamber of deputies vs senate discussion

First, the present work corroborates with previous findings of coalition breakdown, fragmentation, and isolation of Dilma Rousseff's party in the periods preceding the final of the impeachment process [1,11,22,23].

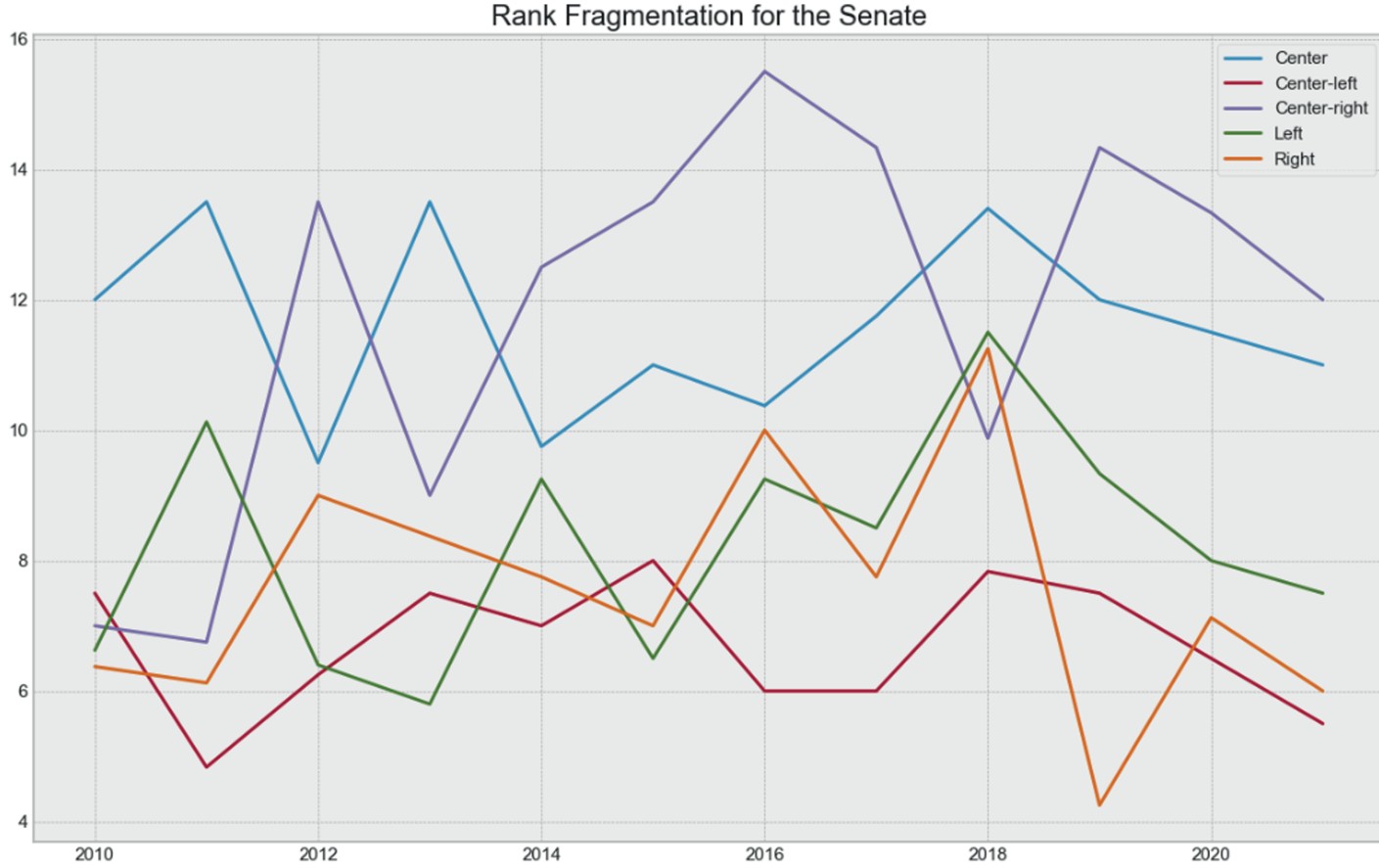

**Fig 20. Senate fragmentation rank by year.**

It also improves over the previous body of work that used backbone extraction methodologies [1,11] to reduce noise in the network by showing that, for the evaluated metrics, there are better algorithms to be applied and that they vary over time. These algorithms achieve higher modularity and Surprise as shown in the Supporting Information.

We also show to be best of our knowledge for the first time, that the Senate behaves differently from the Chamber of Deputies. The reasons for this, besides the clear topological differences of the networks, can be, we hypothesize, because alliances between parties can vary in both houses and the house's president has a high impact on the overall behavior of that house concerning the government.

In fact, it is known that the president of the Chamber of Deputies was highly aggressive towards the government and significantly impacted the impeachment process's success.

However, both houses have the same behavior in terms of ideological communities: their quantity is way smaller than the number of political parties. This is in line with what was found for the Chamber of Deputies by [1,11,23].

## Conclusion and future works

In this article, an analysis of the behavior of both the Brazilian Deputies Chamber and Senate was performed using complex networks, from 2010 to 2021. The aim was to uncover similarities and differences among the party dynamics in both houses. This analysis highlights the

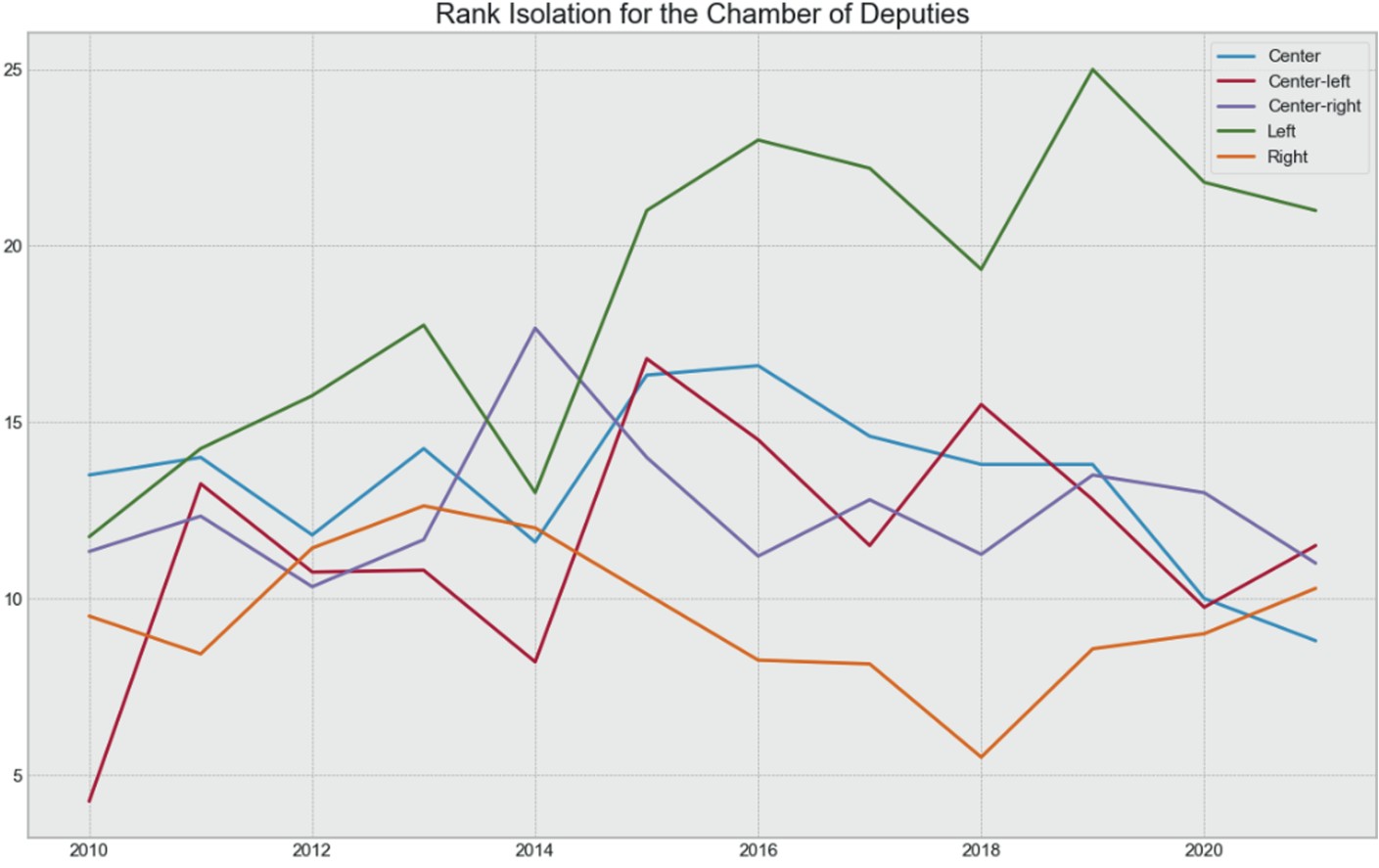

**Fig 21. Chamber of Deputies isolation rank by year.**

following contributions. The first of them is the fact that the Senate was included in this type of analysis of the Brazilian legislative system for the first time.

The second is related to the same methodology previously used to analyze the Chamber of Deputies in several studies can be applicable to Senate and that both houses have different topological structures and group dynamics over time.

The third is that each year has its preferred backbone extraction methodology, and therefore using only one method can reduce the quality of the results. As far as we know, this is also the first time that each year was treated separately in a unified backbone extraction model selection pipeline. We also combine the usage of Modularity and Suprise maximization to deal with shortcomings of both metrics, which, as far as we know, was also done for the first time.

Finally, for the first time, it was verified how communities represent the political spectrum and the government x opposition relationship inside each house, which helped uncover the differences in how the Senate and the Chamber of Deputies faced each of the recent Brazilian political crises.

## Future works

No textual analysis was conducted on the content of the propositions voted on. This is a rich source of information that could be exploited to get more information on how the different parties and communities behave given the theme of the proposition.

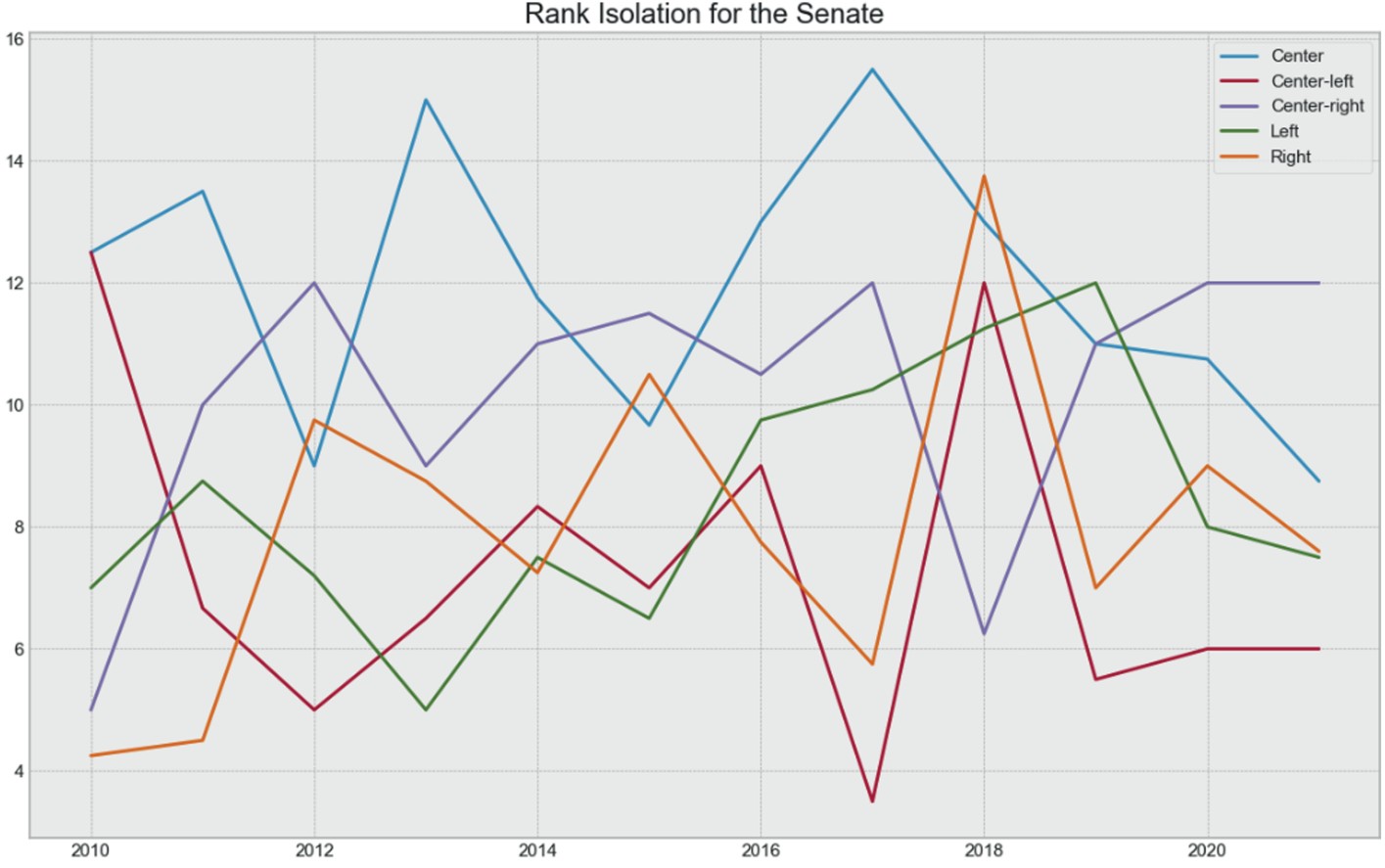

**Fig 22. Senate isolation rank by year.**

Taking into account, the presidents of both houses can also expand this analysis since they are known to exert a great influence on their houses' decisions.

Other clustering methodologies can be used in complex networks but were not considered in the scope of this paper, such as the ones from [38] [39]. These could incorporate the time component directly on the community detection, removing the need to break up the networks by year. Further research could also verify the differences between the non-modularity optimization methods that would be present in the Congress data. Therefore, a clustering consensus algorithm could be applied in case of several discordances.

Finally, the methodology to define the government and opposition groups is somewhat arbitrary, however, it is agnostic to the distribution of the coalition between parties. A future step could be to explore more theory-based cutoff points that do not rely on arbitrary percentile points over the distribution. However, it is important that this new methodology takes into account what was truly done by the parties, not their public intent.

## Supporting information

**S1 Table. Party references.**
(XLSX)

**S1 Fig. Modularity for backbone extraction algorithms for the Chamber of Deputies.**
(TIF)

**S2 Fig. Surprise for backbone extraction algorithms for the Chamber of Deputies.**
(TIF)

**S3 Fig. LCC Size for backbone extraction algorithms for the Chamber of Deputies.**
(TIF)

**S4 Fig. MSE for backbone extraction algorithms for the Chamber of Deputies.**
(TIF)

**S5 Fig. Modularity for backbone extraction algorithms for the Senate.**
(TIF)

**S6 Fig. Surprise for backbone extraction algorithms for the Senate.**
(TIF)

**S7 Fig. LCC Size for backbone extraction algorithms for the Senate.**
(TIF)

**S8 Fig. MSE for backbone extraction algorithms for the Senate.**
(TIF)

**S9 Fig. The political spectrum distribution inside communities for 2013 and 2014.**
(TIF)

**S10 Fig. The political spectrum distribution inside communities for 2015 and 2016.**
(TIF)

**S2 Table. Government and opposition parties for the Chamber of Deputies.**
(XLSX)

**S3 Table. Government and opposition parties for the Senate.**
(XLSX)

## Author contributions

**Conceptualization:** Tiago José de Oliveira Toledo Junior, Roseli Aparecida Francelin Romero.

**Data curation:** Tiago José de Oliveira Toledo Junior.

**Formal analysis:** Tiago José de Oliveira Toledo Junior.

**Funding acquisition:** Diego Raphael Amancio, Roseli Aparecida Francelin Romero.

**Investigation:** Tiago José de Oliveira Toledo Junior.

**Methodology:** Tiago José de Oliveira Toledo Junior, Diego Raphael Amancio.

**Project administration:** Roseli Aparecida Francelin Romero.

**Software:** Tiago José de Oliveira Toledo Junior.

**Supervision:** Diego Raphael Amancio, Roseli Aparecida Francelin Romero.

**Validation:** Roseli Aparecida Francelin Romero.

**Visualization:** Tiago José de Oliveira Toledo Junior.

**Writing – original draft:** Tiago José de Oliveira Toledo Junior.

**Writing – review & editing:** Diego Raphael Amancio, Roseli Aparecida Francelin Romero.

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
