## [Decision Letter · Decision Letter 0]

19 Jul 2024

PONE-D-24-16127Complex Networks Applied to Political Analysis: Group Voting Behavior in the Brazilian CongressPLOS ONE

Dear Dr. Toledo Junior,

Thank you for submitting your manuscript to PLOS ONE. After careful consideration, we feel that it has merit but does not fully meet PLOS ONE’s publication criteria as it currently stands. Therefore, we invite you to submit a revised version of the manuscript that addresses the points raised during the review process.

Please see the comments of two reviewers below. Please note that it is not a requirement to cite any specific works suggested by the reviewers, but please consider their feedback on providing additional context. Please also note that the reviewer comments on the figure quality can be disregarded - the figures are fine when downloaded, which is the resolution that will appear if the manuscript is published. Please also note that you may not be able to rearrange the figures to appear in the correct place in the manuscript for the next revision as requested by one reviewer, but that this will also be resolved by the PLOS systems for eventual publication.

We look forward to receiving your revised manuscript.

Kind regards,

Hanna Landenmark

Staff Editor

PLOS ONE

Journal Requirements:

"Funding provided by São Paulo Research Foundation (FAPESP) for the publication of the research."

Reviewers' comments:

Reviewer's Responses to Questions

**Comments to the Author**

1. Is the manuscript technically sound, and do the data support the conclusions?

Reviewer #1: Yes

Reviewer #2: Partly

2. Has the statistical analysis been performed appropriately and rigorously? 

Reviewer #1: Yes

Reviewer #2: Yes

3. Have the authors made all data underlying the findings in their manuscript fully available?

Reviewer #1: No

Reviewer #2: Yes

4. Is the manuscript presented in an intelligible fashion and written in standard English?

Reviewer #1: Yes

Reviewer #2: Yes

5. Review Comments to the Author

Reviewer #1: The authors analyzed the Brazilian Chamber of Deputies and Senate using complex networks from 2010 to 2021 to uncover similarities and differences between both houses' party dynamics. This manuscript presents interesting results and, therefore, holds the potential for publication after addressing the following points:

Q1: In the introduction, the authors state "The political dynamics of a country can be considered a complex system." This general approach has been recognized for years and should be acknowledged (with references) before focusing on the specific topic addressed by the authors.

Q2: The authors state that "the best algorithm for each year and network is different." However, they do not explain the underlying mechanisms behind such finding. A detailed explanation should be provided in the text.

Q3: The paper's discussion should expand its scope to elucidate the broader significance of the results. The text should address the following questions, supported by references:

a) Exactly which findings corroborate previous results in the literature?

b) How does the complex network analysis presented by the authors complement other methodologies used in the analysis of political data in Brazil?

c) What is the overall picture obtained when integrating the authors' findings with the preexisting body of research on political data in Brazil?

Obs: Several images included in the manuscript are of low resolution (e.g., methodology_overall.png).

Reviewer #2: The article presents a kind of meta analysis of different methodologies to analyze graph topology and detect communities in order to extract information from the Brazilian legislative houses. Though the work done is interesting and may help researchers analyzing similar data, before endorsing it for publication, I fell a few issues should be addressed. I will list below my concerns and observations mostly in the order they appeared while reading the paper.

1) The second item listed in the introduction as specific objectives of the work mentions a community detection methodology already proved to work in the chamber of deputies without specifically mentioning which one they are referring to. I should also mention here that, even though a given methodology does return interesting results about something as subjective as identifying political groups, it should hardly be taken as proof.

2) The list below cites the main contributions of the work. The third one (again mentioning a proof), points to modularity. I will in my comments raise many concerns about the use of this quality function. I suggest the authors use modularity and specially the Louvin algorithm very cautiously and making all important caveats, as this function is full of flaws. I point here a few works on its limitations and alternatives (by DOI): 10.1038/s41598-019-41695-z, 10.1073/pnas.0605965104, 10.1103/PhysRevE.81.046106, 10.1371/journal.pone.0024195, 10.1016/j.physa.2022.127063

3) A note on style: I suggest the authors never start a sentence with a citation as done in session Backbone Extraction Methodologies (shouldn't the sessions be numerated?) where the first paragraph reads: "[18] proposed a framework ...".

4) In the end of this session the authors mention they use Netbone library for the implementation of several methodologies. These several are all the ones used in the present work or others were used as well? This should be precised and if others were used, please mention where the adopted implementation was taken from.

5) I'm not sure on the origin of the problem, but in my version (downloaded from PLOS ONE server), the figures do not appear in the body of the paper, making its reading very difficult. Next time, before the authors approve the submission, they should make sure all figures are visible in their proper places and not only as stand alone images in the end.

6) In the Backbone Extraction session, the authors start by pointing out what is basically a flaw in the modularity/Louvin algorithm: Namely, that the modularity does not depend solely in the local topological characteristics of the communities, but on the whole network, making the same local topology resulting in different community structures depending on the size of the graph as a whole. Again: be aware of the limitations of Louvin and modularity and how it may impact its use for a specific goal.

7) In Contextual Analysis, I could not really understand what was done. Please, improve the description of the problem at hand you are trying to solve and the solution adopted. Finally here, in the list, I see again that a maximization of modularity is sought. Another important flaw of Q, that may directly hinder its use to divide a parliament into political groups, is the fact that it is biased toward dividing the graph into communities of similar sizes (10.1016/j.physa.2022.127063). There is no reason why all political parties should be of similar sizes.

8) In session Government and Opposition the way the communities are chosen as either government and opposition seems fairly arbitrary. Moreover, if the same is done for the chamber of deputies and senate, there is a big problem as the senate is smaller and has, typically, less parties in its composition. For instance in the next session it is mentioned that the senate has typically 9 parties, how are they then divided into government, opposition and neutral if the 5 closest to the government party are the government and the 5 furthest the opposition?

9) There are a few instances where a Table is presented as a figure (which I cannot read in my version of the manuscript). Please, make tables as tables, not as figures.

10) In session Network Visualization, the last paragraph states that using the same algorithm to both houses can be misleading without some kind of performance test. Well, the thing is that a test can, as well, be biased or subjective, specially if it is modularity based. I would be always inclined to find an algorithm that does work in both cases if I am to compare them both, pointing, whenever the case, to possible caveats. Using many methodologies and selecting the best for each case is a kind of meta analysis of data.

11) The next two sessions touch on the subjective matter of political spectrum and parties are labeled as left, right, center, ... Please, state the criteria adopted for these assignments. This is a subjective matter and even though a party may claim to be left or right inclined, once one looks at how their congressmen voted, in most cases they are not really voting coherently in a given subject and may even be divided among communities associated to government and opposition.

12) There are other works that analyse the Brazilian congress, in particular the impeachment period the authors mentioned and analyse, some even pointing to similar conclusions. Please, take a look at 10.1371/journal.pone.0226504 and 10.1016/j.physa.2021.125832, for example.

6. PLOS authors have the option to publish the peer review history of their article (what does this mean?). If published, this will include your full peer review and any attached files.

Reviewer #1: No

Reviewer #2: **Yes: **Daniel Gamermann

---

## [Author Response · Author response to Decision Letter 1]

4 Sep 2024

I've attached the rebuttal letter in the files of the submission.

---

## [Decision Letter · Decision Letter 1]

8 Oct 2024

PONE-D-24-16127R1Complex Networks Applied to Political Analysis: Group Voting Behavior in the Brazilian CongressPLOS ONE

Dear Dr. Toledo Junior,

Thank you for submitting your manuscript to PLOS ONE. After careful consideration, we feel that it has merit but does not fully meet PLOS ONE’s publication criteria as it currently stands. Therefore, we invite you to submit a revised version of the manuscript that addresses the points raised during the review process.

Please submit your revised manuscript by Nov 22 2024 11:59PM. If you need more time than this to complete your revisions, please reply to this message or contact the journal office at plosone@plos.org. Please include the following items when submitting your revised manuscript:

We look forward to receiving your revised manuscript.

Kind regards,

Dragana Bozic Lenard

Academic Editor

PLOS ONE

Reviewers' comments:

Reviewer's Responses to Questions

**Comments to the Author**

1. If the authors have adequately addressed your comments raised in a previous round of review and you feel that this manuscript is now acceptable for publication, you may indicate that here to bypass the “Comments to the Author” section, enter your conflict of interest statement in the “Confidential to Editor” section, and submit your "Accept" recommendation.

Reviewer #1: All comments have been addressed

Reviewer #2: All comments have been addressed

Reviewer #3: (No Response)

2. Is the manuscript technically sound, and do the data support the conclusions?

Reviewer #1: Yes

Reviewer #2: Yes

Reviewer #3: No

3. Has the statistical analysis been performed appropriately and rigorously? 

Reviewer #1: Yes

Reviewer #2: Yes

Reviewer #3: No

4. Have the authors made all data underlying the findings in their manuscript fully available?

Reviewer #1: Yes

Reviewer #2: Yes

Reviewer #3: Yes

5. Is the manuscript presented in an intelligible fashion and written in standard English?

Reviewer #1: (No Response)

Reviewer #2: Yes

Reviewer #3: Yes

6. Review Comments to the Author

Reviewer #1: (No Response)

Reviewer #2: The authors have addressed my concerns in the revised version. I'm sorry about my comment that some tables appeared as figures. I mistook Fig. 5 for a table.

Reviewer #3: # Review

The paper studies how backbone extraction and community detection can provide useful information on voting behaviour in the Brazilian Congress (Deputies and Senate). My knowledge about the Brazilian Congress is limited, but I do have knowledge about the methods involved, and so will restrict my review to that aspect.

I do not understand the reason why a backbone extraction is performed as a sort of preprocessing step for the community detection, and it makes little sense to me. There is absolutely no problem in running community detection algorithms on the type of provided networks, and there are many other potential algorithms that could be tried, without resorting to first performing a backbone extraction.

For example, Mucha et. al (2010) proposed a temporal variation of community detection, and also applied it to voting networks in parliament. That approach seems quite well the application of the authors, and is a more reasonable way of getting rid of the “noisy edges” (i.e. some substantial difference need to arise in order to reclassify nodes from one cluster to another, thus reducing the “noise”). Why not use that type of approach? Alternatively, approaches such as Peixoto & Rosvall (2015) seem quite applicable to the type of temporal variation that is of interest to the researchers.

Moreover, the evaluation of what “works” best is rather shallow. That is, the evaluation of the methodology by the provided metrics does not really clarify much. Nor is the evaluation of the backbone extraction by the maximum modularity very informative. An “expert-based” evaluation of the quality of a partition might be more informative, but this is of course also rather challenging. But why should a backbone extraction approach be better if it results in a higher modularity? This remains unclear to me.

Finally, the authors train a machine learning (ML) model for predicting edge weights (see p. 8, l. 279), but it is not clear to me where this suddenly comes from, nor what the role of that ML model is in the remainder of the paper. Why is this used? What is this used for? How does it relate to the community detection approach?

Overall then, I’m afraid that I don’t see what the contribution of this paper is. The technical contribution is rather limited and not well motivated. The social scientific contribution about the Brazilian parliament may be more relevant, but given the large role of the technical contribution in the paper, the social scientific contribution seems more marginal. For that reason, I would recommend to reject the paper. It seems that this approach (backbone -> community detection) has been used before in the same context, and I have my questions about that approach more generally. But even if that approach might make sense, it is not clear to me how the current manuscript shows that the other approaches that the authors consider (i.e other combinations of backbone / community detection) are “better”, and for what reason.

Minor details:

- p. 5, why only keep edges with weight > 0? The negative edges also provide quite some information about relevant differences, and could inform community detection approaches.

- p. 6, the section on “metric selection for backbone extraction” directly starts with metrics to evaluate the quality of communities, which is rather strange. Please first introduce the backbone extraction itself, and why you end up using community detection methods (which is not clear to me at all)

- p. 9, the “government” and “opposition” is defined in terms of percentiles. However, it is not clear what percentiles are being calculated, that is on what measure is the distribution (and percentiles) defined? Moreover, this approach does not really seem to make sense to me, but perhaps there are no clear governing coalitions defined in Brazilian politics, such that another approach is necessary. Without further  argumentation this is not clear.

# References

Mucha, P. J., Richardson, T., Macon, K., Porter, M. A., & Onnela, J.-P. (2010). Community structure in time-dependent, multiscale, and multiplex networks. *Science (New York, N.Y.)*, *328*(5980), 876–878. [https://doi.org/10.1126/science.1184819](https://doi.org/10.1126/science.1184819)

Peixoto, T. P., & Rosvall, M. (2015). Modeling sequences and temporal networks with dynamic community structures. *Nature Communications 2017 8:1*, *8*(1), 5. [https://doi.org/10.1038/s41467-017-00148-9](https://doi.org/10.1038/s41467-017-00148-9)

7. PLOS authors have the option to publish the peer review history of their article (what does this mean?). If published, this will include your full peer review and any attached files.

Reviewer #1: No

Reviewer #2: **Yes: **Daniel Gamermann

Reviewer #3: No

---

## [Author Response · Author response to Decision Letter 2]

19 Dec 2024

The answers are available on the sent file.

---

## [Decision Letter · Decision Letter 2]

6 Feb 2025

Complex Networks Applied to Political Analysis: Group Voting Behavior in the Brazilian Congress

PONE-D-24-16127R2

Dear author,

We’re pleased to inform you that your manuscript has been judged scientifically suitable for publication and will be formally accepted for publication once it meets all outstanding technical requirements.

Kind regards,

Dragana Bozic Lenard

Academic Editor

PLOS ONE

Additional Editor Comments (optional):

Reviewers' comments:

Reviewer's Responses to Questions

**Comments to the Author**

1. If the authors have adequately addressed your comments raised in a previous round of review and you feel that this manuscript is now acceptable for publication, you may indicate that here to bypass the “Comments to the Author” section, enter your conflict of interest statement in the “Confidential to Editor” section, and submit your "Accept" recommendation.

Reviewer #1: All comments have been addressed

Reviewer #2: All comments have been addressed

2. Is the manuscript technically sound, and do the data support the conclusions?

Reviewer #1: Yes

Reviewer #2: Yes

3. Has the statistical analysis been performed appropriately and rigorously? 

Reviewer #1: Yes

Reviewer #2: Yes

4. Have the authors made all data underlying the findings in their manuscript fully available?

Reviewer #1: Yes

Reviewer #2: Yes

5. Is the manuscript presented in an intelligible fashion and written in standard English?

Reviewer #1: Yes

Reviewer #2: Yes

6. Review Comments to the Author

Reviewer #1: I have reviewed the improvements made by the authors, and I find that the manuscript is now of sufficient quality for publication.

Reviewer #2: In the my previous review I had already accepted the paper.

I had already seen my concerns answered in the previous version of the paper.

7. PLOS authors have the option to publish the peer review history of their article (what does this mean?). If published, this will include your full peer review and any attached files.

Reviewer #1: No

Reviewer #2: No

---

## [Editor Report · Acceptance letter]

PONE-D-24-16127R2

PLOS ONE

Dear Dr. Toledo Junior,

I'm pleased to inform you that your manuscript has been deemed suitable for publication in PLOS ONE. Congratulations! Your manuscript is now being handed over to our production team.

Kind regards,

on behalf of

Dr. Dragana Bozic Lenard

Academic Editor

PLOS ONE